# Corrective feedback guides human perceptual decision-making by informing about the world state rather than rewarding its choice

**Hyang-Jung Lee**[1], **Heeseung Lee**[1], **Chae Young Lim**[2], **Issac Rhim**[3], **Sang-Hun Lee**[1]*

**1** Department of Brain and Cognitive Sciences, Seoul National University, Seoul, South Korea, **2** Department of Statistics, Seoul National University, Seoul, South Korea, **3** Institute of Neuroscience, University of Oregon, Eugene, Oregon, United States of America

* visionsl@snu.ac.kr

## Abstract

Corrective feedback received on perceptual decisions is crucial for adjusting decision-making strategies to improve future choices. However, its complex interaction with other decision components, such as previous stimuli and choices, challenges a principled account of how it shapes subsequent decisions. One popular approach, based on animal behavior and extended to human perceptual decision-making, employs "reinforcement learning," a principle proven successful in reward-based decision-making. The core idea behind this approach is that decision-makers, although engaged in a perceptual task, treat corrective feedback as rewards from which they learn choice values. Here, we explore an alternative idea, which is that humans consider corrective feedback on perceptual decisions as evidence of the actual state of the world rather than as rewards for their choices. By implementing these "feedback-as-reward" and "feedback-as-evidence" hypotheses on a shared learning platform, we show that the latter outperforms the former in explaining how corrective feedback adjusts the decision-making strategy along with past stimuli and choices. Our work suggests that humans learn about what has happened in their environment rather than the values of their own choices through corrective feedback during perceptual decision-making.

## Introduction

Perceptual decision-making (PDM) means committing to a proposition about an objective world state (e.g., "The temperature today is low."). Decision-makers adjust future commitments based on what they experienced from past commitments, including what they perceived, what they chose, and what the environment gave them in return. Among these history factors, *trial-to-trial corrective feedback*—feedback about the correctness of a decision-maker's choices on a trial-to-trial basis—is widely used by experimenters to train participants on PDM tasks. Despite this clear utility of feedback and a pile of evidence for its impact on subsequent PDM behavior across species and sensory modalities [1–11], much remains elusive about how corrective feedback, in conjunction with other history factors, exerts its trial-to-trial influence on subsequent decisions.

**Data Availability Statement:** Raw data, processed data files, and codes used in this study are publicly available on GitHub at: https://github.com/hyangjung-lee/lee_2023_corrective-fbk (https://doi.

org/10.5281/zenodo.8427373). All other relevant data, including numerical data underlying each figure, are within the paper and its Supporting Information files.

**Funding:** This research was supported by the Seoul National University (SNU) Research Grant 339-20220013 (to S.-H. L.), by the Brain Research Program through the National Research Foundation of Korea (NRF) funded by the Ministry of Science and Information and Communications Technology (MSIT) Grant No. NRF-2021R1F1A1052020 (to S.-H. L.), and by the Basic Research Laboratory Program through NRF funded by MSIT Grant No. NRF-2018R1A4A1025891 (to S.-H. L.). The funders had no role in study design, data collection, analysis, the decision to publish, or manuscript preparation.

**Competing interests:** The authors have declared that no competing interests exist.

**Abbreviations:** AICc, Akaike information criterion corrected; BADS, Bayesian Adaptive Direct Search; BDT, Bayesian Decision Theory; BMBU, Bayesian model of boundary-updating; DVA, degree in visual angle; IBS, inverse binomial sampling; PDM, perceptual decision-making; PSE, point of subjective equality; RL, reinforcement learning; *toi*, trial of interest; VDM, value-based decision-making.

Unlike PDM, value-based decision-making (VDM) involves making choices based on decision-makers' subjective preferences (e.g., "choosing between two drinks based on their tastes"). Reinforcement learning (RL) algorithms have proven effective in explaining how past rewards affect future VDM based on error-driven incremental mechanisms [12–18]. Intriguingly, there have been attempts to explain the impact of past feedback on subsequent PDM by grafting an RL algorithm onto the PDM processes [3,4,8–10]. This grafting premises that decision-makers treat corrective feedback in PDM similarly to reward feedback in VDM. On this premise, this RL-grafting account proposes that decision-makers update the *value* of their choice to minimize the difference between the expected reward and the actual reward received, called "reward prediction error" (red dashed arrows in Fig 1A). Importantly, the amount of reward prediction error is inversely related to the strength of sensory evidence—i.e., the extent to which a given sensory measurement of the stimulus supports the choice—because the expected

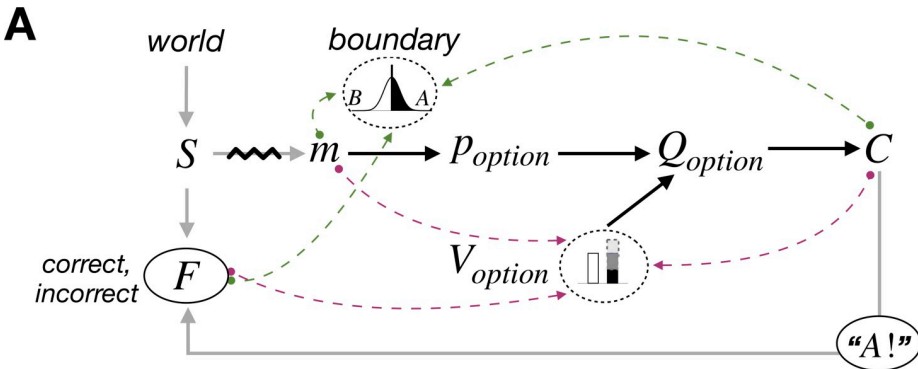

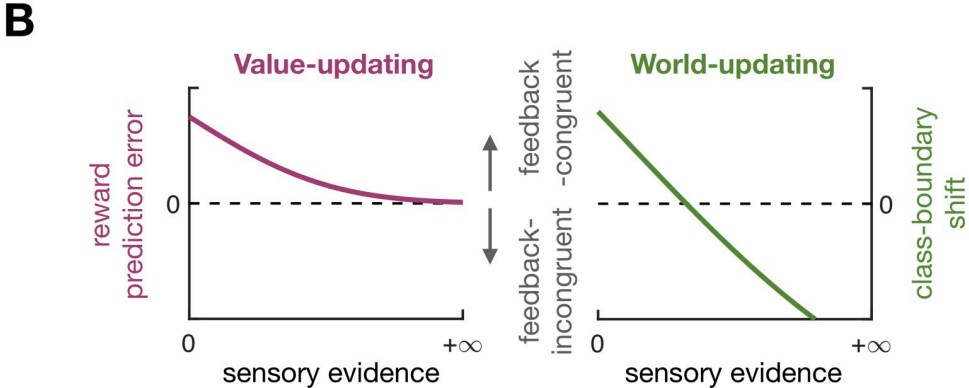

**Fig 1. Two possible scenarios for what humans learn from feedback for PDM and their distinct predictions of feedback effects. (A)** Decision-making platform for perceptual binary classification. The gray arrows depict how a sensory measurement $m$ and feedback $F$ are generated from a stimulus $S$, which is sampled from the *world*, and a choice $C$. The black arrows depict the computational process, where, for a given choice *option*, a decision-maker computes its expected value $Q_{option}$ by multiplying the probability that the choice is correct $p_{option}$ given $m$ and the class boundary $B$ with the value of that choice $V_{option}$ and make a choice $C$ based on $Q_{option}$. In principle, the decision-maker may update either $V_{option}$ (red dashed arrows; value-updating) or *world* (green dashed arrows; world-updating) from $m$, $C$, and $F$. **(B)** Distinct sensory evidence–dependent feedback effects predicted by the value-updating and world-updating scenarios. According to the value-updating scenario (left), as sensory evidence becomes stronger, $p_{option}$ increases, and accordingly, so does $Q_{option}$. As a result, reward prediction errors become smaller but remain in the direction congruent with feedback, which predicts that feedback effects on subsequent trials diminish asymptotically as a function of the strength of sensory evidence. According to the world-updating scenario (right), as sensory evidence becomes stronger, the stimulus distribution, and accordingly $B$ too, becomes shifted farther towards the stimulus in the direction counteracting the influence of feedback. As a result, the direction of feedback effects is the same as that predicted by the value-updating scenario for weak sensory evidence but eventually reverses to the direction incongruent with feedback as sensory evidence becomes stronger.

value becomes low as the sensory evidence becomes weak. For example, suppose a decision-maker committed to a proposition, "The temperature today is low." Then, *correct* feedback to that commitment increases the value of the "low" choice since the positive reward for the "low" choice leads to the positive reward prediction error, which indicates the need to heighten the value of the "low" choice. Importantly, the amount of value-updating is greater when the experienced temperature is moderately cold (e.g., −2˚C, weak sensory evidence for the "low" choice) compared to when it is very cold (e.g., −15˚C, strong sensory evidence for the "low" choice) because the expected reward is smaller in the former, which leads to a greater level of reward prediction error compared to the latter (as illustrated in the left panel of Fig 1B). A recent study [9] referred to this sensory evidence–dependent impact of feedback as "confidence-guided choice updating" based on the tight linkage between decision confidence and sensory evidence. This RL-grafting account, referred to as the *value-updating scenario* hereinafter, appears natural given that corrective feedback is typically provided as physical rewards such as juice or water in animal PDM experiments [4,5,8–10,19–21]. The value-updating scenario seems plausible from the perspective that PDM and VDM might share common mechanisms [22], as suggested by some empirical studies [23,24].

Nevertheless, value-updating might not be the only route through which feedback effects transpire in PDM, especially for humans receiving corrective feedback without any physical rewards. Alternatively, decision-makers may treat feedback not as rewards but as a logical indicator of whether the proposition they committed to is true or false in the world. In this scenario, decision-makers update their belief about world statistics (i.e., stimulus distribution) by combining the information about the trueness of their choice, which is informed by feedback, and the information about the stimulus, which is informed by a sensory measurement (dashed arrow from *m* in Fig 1A). Suppose you have recently arrived in Canada for the first time in the winter and felt the chilly air. You remarked, "The temperature today is low." Your friend, who has lived for long in Canada, may agree or disagree with you, and this will provide you with information on the typical temperature distribution during the Canadian winter. The *incorrect* feedback from your friend (e.g., "Actually, it's not low at all today.") indicates that the temperature experienced today falls on the higher side of the actual distribution, making you adjust your belief about the distribution to the lower side. On the contrary, the *correct* feedback (e.g., "Yes, it's low today.") will lead you to adjust your belief about the distribution to the higher side. It is important to note that, besides the feedback from your friend, the temperature felt by yourself also informs you of the statistical distribution of temperature since it is a sample from that distribution. For example, if the temperature felt moderately cold (e.g., −2˚C), your belief about the temperature distribution will only slightly shift towards the lower side. However, if it felt very cold (e.g., −15˚C), your belief will shift towards the same lower side, but with a much greater amount, which can counteract the impact of the *correct* feedback on your belief (i.e., adjusting your belief to the higher side).

Therefore, according to this alternative scenario, referred to as the *word-updating scenario* hereinafter, *correct* feedback to "The temperature today is low." will increase the tendency to classify the next day's temperature as "low," just like the value-updating scenario. However, unlike the value-updating scenario, the world-updating scenario implies that when sensory evidence is too strong, such a tendency can be reversed, leading to a counterintuitive increase in the tendency to classify the next day's temperature as "high," (as illustrated in the right panel of Fig 1B). The world-updating scenario is conceptually parsimonious because it does not require any component outside the PDM processes, such as the RL algorithms developed in the VDM. Especially in Bayesian Decision Theory (BDT) [25,26], which has been providing compelling accounts for PDM behavior, world statistics is a crucial knowledge that is required to infer a world state in PDM [27–30].

Here, we tested which of the 2 scenarios better explains the effects of corrective feedback—without any physical reward—on humans' PDM. To do so, we implemented the value-updating and world-updating scenarios into a variant of RL model [9] and a Bayesian model, respectively, and directly compared the 2 models' accountability for the feedback effects on humans' PDM behavior. As a PDM task, we opted for a binary classification task, one most widely used PDM task in which decision-makers sort items into 2 discrete classes by setting a boundary since the 2 scenarios make distinct predictions about the stimulus-dependent feedback effects in this task. As was described intuitively above and will be explained rigorously later, the value-updating scenario predicts that feedback, which acts like rewards, "unidirectionally" fosters and suppresses the rewarded (*correct*) and unrewarded (*incorrect*) choices, respectively, in subsequent trials while diminishing its impact asymptotically as sensory evidence becomes stronger, due to the reduction in reward prediction error (the red curve in Fig 1B). By contrast, the world-updating scenario predicts that the feedback effects not just diminish but eventually become reversed to the opposite side as sensory evidence becomes stronger, as the shift of the class boundary towards the previous stimulus counteracts the boundary shift due to feedback (the green curve in Fig 1B).

We found the world-updating model superior to the value-updating model in explaining human history effects of corrective feedback on PDM. Critically, the value-updating model fails to account for the observed stimulus-dependent feedback effects. Our findings suggest that humans are likely to treat corrective feedback in PDM as logical indicators of the trueness of the proposition to which they committed, rather than as rewards, and update their knowledge of world statistics, rather than the values of their choices, based on feedback in conjunction with the other history factors—previous stimuli and choices.

## Results

### Quantifying the retrospective and prospective history effects of feedback on binary classification

To study the stimulus-dependent feedback effects in PDM, we acquired long sequences (170 trials/sequence) of binary choices ($C \in \{small, large\}$) many times (30 sequences/participant) from each of 30 human participants while varying the ring size ($S \in \{-2, -1, 0, 1, 2\}$) and providing corrective feedback ($F \in \{correct, incorrect\}$) (Fig 2A). On each trial, participants viewed a ring, judged whether its size is *small* or *large* as accurately as possible while receiving feedback, which indicated by color whether the choice was correct or incorrect (Fig 2B). We ensured the ring size varied sufficiently—including the ones very easy and difficult for classification—so that the 2 scenarios' distinct predictions on the stimulus-dependent feedback effects could be readily compared. Also, we used stochastic feedback, where *correct* and *incorrect* feedback was occasionally given to incorrect and correct choices, respectively, to cover the entire 3D space of decision-making episodes defined orthogonally over "stimulus," "choice," and "feedback" (5×2×2 = 20 episodes; Fig 2C; Materials and methods).

To rigorously evaluate the correspondence between model prediction and human behavior, we quantified the history effects in both retrospective and prospective directions of time, as follows (Fig 2D). First, we localized the trials in which a PDM episode of interest occurred (trial of interest, *toi*) and stacked the trials that preceded (the retrospective block of trials, *toi*−1) and those that followed (the prospective block of trials, *toi*+1) the *toi*. Second, we derived the 2 psychometric curves from the retrospective and prospective blocks of trials, respectively, and fit the cumulative normal distribution function to these curves to estimate the point of subjective equality (PSE) measures, which have previously been used [19–21] and known to reliably estimate the history-dependent choice biases in PDM [31]. Thus, the PSEs of the retrospective

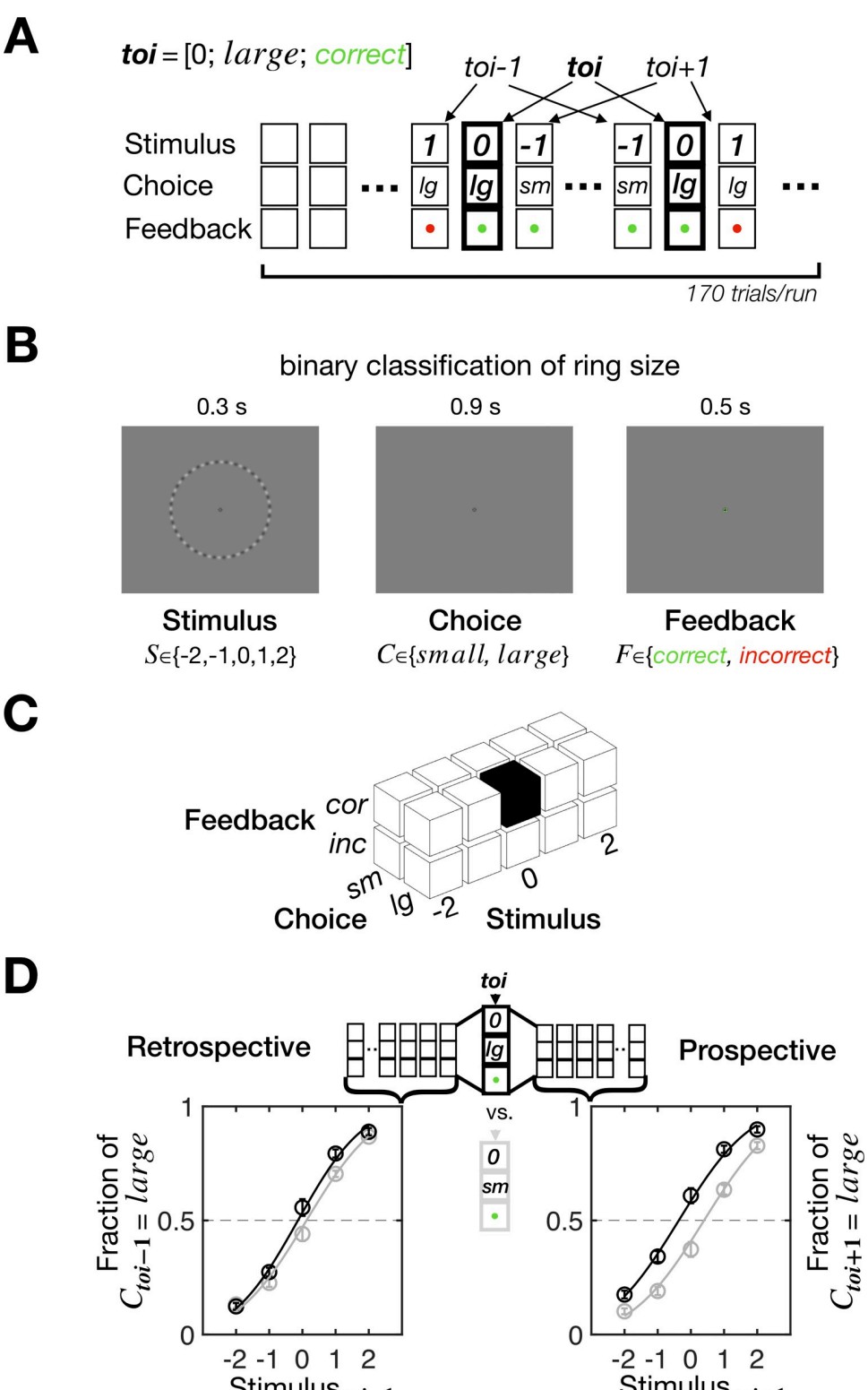

**Fig 2. Experimental design and definition of retrospective and prospective history effects. (A)** A chain of PDM episodes over a single sequence of trials. Each trial sequence consists of 170 column vectors of PDM episode [stimulus; choice; feedback]. In this example, the trial of interest (*toi*) is characterized by an episode vector [0; *large*; *correct*] and demarcated by thick outlines. The trials that precede and follow *toi* can be labeled as *toi*−1 and *toi*+1, respectively. **(B)** Trial structure. Participants viewed a randomly sampled ring with their eyes fixed, classified its size, and then received

feedback indicating whether the classification was correct or incorrect by the color around the fixation. **(C)** The 3D state space of the PDM episodes in the experiment. The example episode of *toi* in **(A)** is marked by the black cube. **(D)** Definition of retrospective and prospective history effects. As illustrated in **(A)** and **(C)**, for any given episode of *toi*, all the trials labeled with *toi*−1 and *toi*+1 are stacked and used to derive the psychometric curves, respectively. The PSEs estimated for the *toi*−1 and *toi*+1 psychometric curves quantify the retrospective and prospective history effects, respectively. In this example, the black and gray curves were defined for *toi* = [0; *large*; *correct*] and *toi* = [0; *small*; *correct*], respectively, with circles and bars representing the mean and SEM across 30 participants, respectively. The data underlying this figure **(D)** can be found in S1 Data.

and prospective trials quantify the choice biases that exist *before* and *after* the PDM episode of interest occurs, respectively, with negative and positive values signifying that choices are biased to *large* and *small*, respectively.

## Decision-making processes for binary classification

As a first step of evaluating the value-updating and world-updating scenarios, we constructed a common platform of decision-making for binary classification where both scenarios play out. This platform consists of 3 processing stages (Fig 3A). At the stage of "perception," the decision-maker infers the class probabilities, i.e., the probabilities that the ring size ($S$) is larger and smaller, respectively, than the class boundary ($B$) given a noisy sensory measurement ($m$), as follows:

$$p(CL = large) = p(S > B|m) = \int_{B}^{\infty} p(S|m)dS;$$

$$p(CL = small) = 1 - p(CL = large),$$

where $CL$ stands for the class variable with the 2 (*small* and *large*) states.

At the stage of "valuation," the decision-maker forms the expected values for the 2 choices ($Q_{large}$ and $Q_{small}$) by multiplying the class probabilities by the learned values of the corresponding choices ($V_{large}$ and $V_{small}$) as follows:

$$Q_{large} = p(CL = large) \times V_{large};$$

$$Q_{small} = p(CL = small) \times V_{small}.$$

Lastly, at the stage of "decision," the decision-maker commits to the choice whose expected value is greater than the other. In this platform, choice bias may originate from the perception or valuation stage. Suppose the decision-maker's belief about size distribution at the perception stage is not fixed but changes depending on previous PDM episodes (Fig 3B, top). Such changes lead to the changes in PSE of the psychometric curve because the class probabilities change as the class boundary changes (Fig 3B, bottom). Alternatively, suppose the decision-maker's learned values of the choices are not fixed but change similarly (Fig 3C, top). These changes also lead to the changes in PSE of the psychometric curve because the expected values change as the choice values change (Fig 3C, bottom).

## The belief-based RL model

To implement the value-updating scenario, we adapted the belief-based RL model [9] to the current experimental setup. Here, feedback acts like a reward by positively or negatively reinforcing the value of choice ($V_{large(small)}$) with the deviation of the reward outcome ($r$) from the expected value of that choice ($Q_{large(small)}$), as follows:

$$V_{large(small)} \leftarrow V_{large(small)} + \alpha\delta;$$

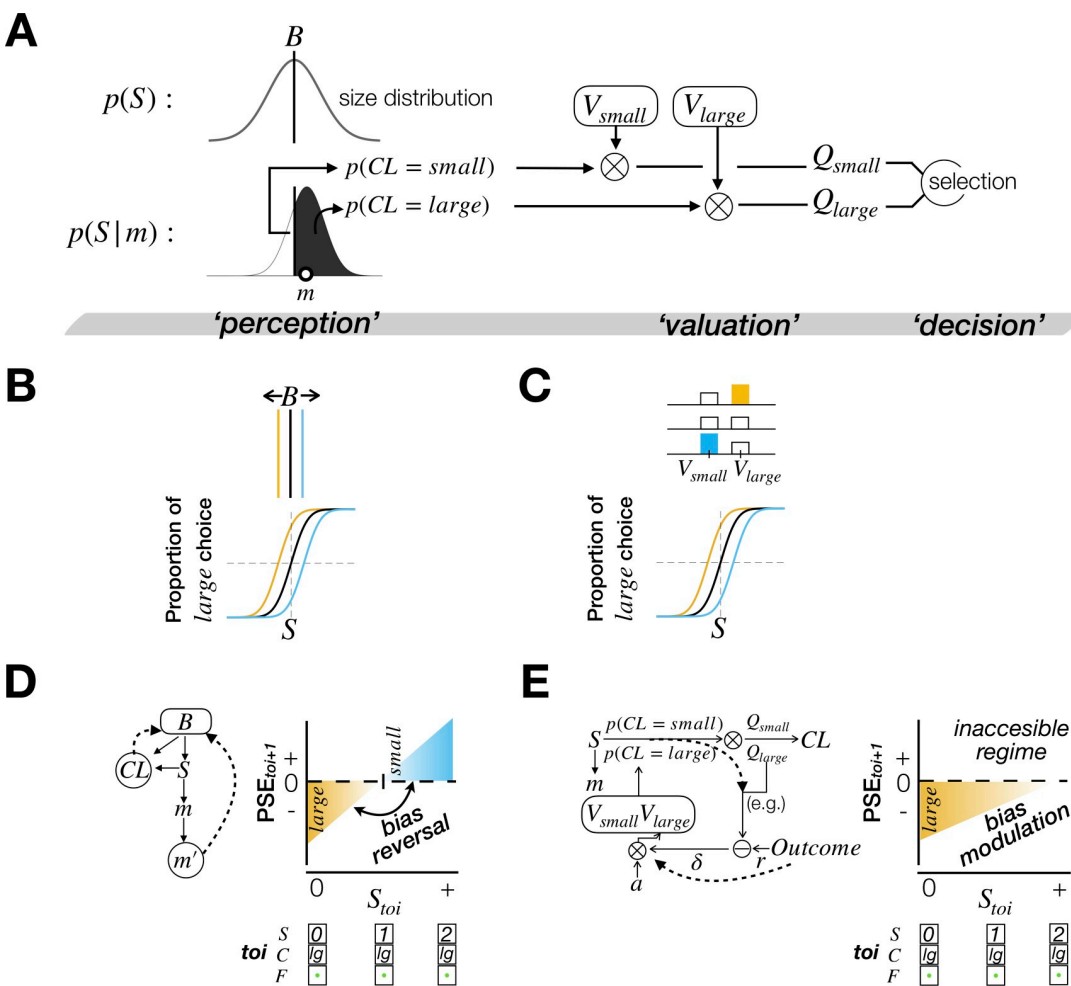

**Fig 3. Implementation of the value-updating and world-updating scenarios into computational models in a common PDM platform. (A)** Computational elements along the 3 stages of PDM for binary classification. At the "*perception*" stage, the probabilities that the class variable takes its binary states *small* and *large*—$p(CL = large)$ and $p(CL = small)$—are computed by comparing the belief on the stimulus size $p(S|m)$ against the belief on the class boundary $B$—the mean of the belief on stimulus distribution in the world $p(S)$. At the "*valuation*" stage, the outcomes of the perception stage are multiplied by the learned values $V$s to produce the expected values $Q$s. At the "*decision*" stage, the choice with the greater expected value is selected. **(B, C)** Illustration of 2 potential origins of choice biases, one at the "*perception*" stage **(B)** and the other at the "*valuation*" stage **(C)**. The color indicates the direction of choice bias (yellow for bias to *large*; black for no bias; blue for bias to *small*). **(D, E)** Illustration of the architectures (left panels) and predictions on the stimulus-dependent feedback effects (right panels) of BMBU **(D)** and the belief-based RL model **(E)**. In the left panels, the dashed arrows represent the ways the history factors (feedback and stimulus) exert their contribution to choice bias. In the right panels, $PSE_{toi+1}$, which quantifies the choice bias in the trials following a certain PDM episode at $toi = [0; large; correct]$, is plotted as a function of the stimulus size at *toi*. The color indicates the direction of choice bias, as in **(B)** and **(C)**.

$$\delta = r - Q_{large(small)} = r - p(CL = large(small)) \times V_{large(small)},$$

where $\alpha$, $\delta$, and $r$ are the learning rate, the reward prediction error, and the reward, respectively. The state of feedback determines the value of $r$: $r = 1$ for *correct*; $r = 0$ for *incorrect*. Note that $\delta$ has the statistical decision confidence at the perception stage, i.e., $p(CL = large(small))$, as one of its 3 arguments. As stressed by the authors who developed this algorithm [9], this feature makes the strength of sensory evidence—i.e., statistical decision confidence—modulate

the degree to which the decision-maker updates the chosen value based on feedback (Fig 3E, left). Hence, this belief (confidence)-based modulation of value-updating underlies the stimulus-dependent feedback effects: The amount of feedback effects decreases as sensory evidence becomes stronger since the reward prediction error decreases as a function of $p(CL = large$ (*small*)), which is proportional to sensory evidence (Fig 3E, right).

## The Bayesian model of boundary-updating (BMBU)

To implement the world-updating scenario, we developed BMBU, which updates the class boundary based on the previous PDM episode in the framework of BDT. Specifically, given "a state of the class variable that is indicated jointly by feedback and choice," *CL*, and "a noisy memory recall of the sensory measurement (which will be referred to as 'mnemonic measurement' hereinafter)," $m'$, BMBU infers the mean of the size distribution (i.e., class boundary), *B*, by updating its prior belief about *B*, $p(B)$, with the likelihood of *B*, $p(m', CL|B)$, by inverting its learned generative model of how $m'$ and *CL* are generated (Fig 3D, left; Eqs 3–6 in Materials and methods for the detailed formalisms for the learned generative model), as follows:

$$p(B|m', CL) \propto p(m', CL|B)p(B) \equiv p(m', C, F|B)p(B).$$

This inference uses multiple pieces of information from the PDM episode just experienced, including the mnemonic measurement, choice, and feedback, to update the belief about the location of the class boundary (refer to Eqs 8–14 in Materials and methods for more detailed formalisms for the inference). In what follows, we will explain why and how this inference leads to the specific stimulus-dependent feedback effects predicted by the world-updating scenario (Fig 3D, right), where world knowledge is continuously updated.

Suppose a decision-maker currently believes that the size distribution is centered around 0. Let us first consider a case where the decision-maker experiences a PDM episode with an ambiguous stimulus: The ring with size 0 is presented and produces a sensory measurement *m* that is only slightly greater than 0 (through the stochastic process where *m* is generated from *S*; Eq 5), which leads to the *large* choice since the inferred *S* from such *m* is greater than the center of the size distribution (Eqs 4 and 7), and then followed by *correct* feedback. BMBU predicts that after this PDM episode, the decision-maker will update the belief about the size distribution by shifting it towards the smaller side. Hence, the choice in the next trial will be biased towards the larger option, resulting in a negatively biased PSE for the psychometric curve defined by the trials following the episode of interest. This is because the impact of the mnemonic measurement on boundary-updating is minimal, whereas that of the informed class variable is substantial. After the above episode, the decision-maker's noisy mnemonic measurement $m'$ is also likely to be slightly larger than 0 since $m'$ is an unbiased random sample of the sensory measurement *m* (Eq 6). Thus, the impact of $m'$ on boundary updating is minimal because $m'$ is close to 0 and thus only slightly attracts the class boundary. On the contrary, the impact of the informed state of the class variable *CL* on boundary updating is relatively substantial, pushing the class boundary towards the regime consistent with the informed state of *CL* (Eqs 9–12), which is the smaller side. As a result, the class boundary is negatively (towards-*small*-side) biased, which leads to the negative bias in the PSE of the psychometric curve defined from the trials following the episode of interest (as depicted by the left (yellow) regime in the plot of Fig 3D).

Next, to appreciate the stimulus-dependent nature of feedback effects in the world-updating scenario, let us consider another case where the decision-maker experiences a PDM episode with an unambiguous stimulus: The ring with size 2 is presented and produces a sensory measurement *m* that falls around 2, which leads to the *large* choice and then followed by *correct*

feedback. After this episode, as in the previous case with an ambiguous stimulus, the informed state of the class variable ($CL = large$) shifts the class boundary to the smaller side. However, unlike the previous case, the impact of the mnemonic measurement $m'$ on boundary-updating, which is likely to be around 2, is substantial, resulting in a shift of the boundary towards the far larger side. Consequently, the class boundary becomes positively (towards-*large*-side) biased. Here, the mnemonic measurement and the informed state of the class variable exert conflicting influences on boundary updating. Since the mnemonic measurement increases as the stimulus size grows (e.g., $S = 0 \rightarrow 1 \rightarrow 2$), the relative impact of the mnemonic measurement on boundary-updating is increasingly greater as the stimulus size grows, eventually overcoming the counteracting influence of the informed state of the class variable (S1 Fig). As a result, the bias in the class boundary is initially negative but is progressively reversed to be positive as the stimulus size grows, which leads to the bias reversal in the PSE of the psychometric curve defined from the trials following the episode of interest (as depicted by the right (blue) regime in the plot of Fig 3D).

We stress that this "stimulus-dependent bias reversal" is a hallmark of the world-updating scenario's prediction of the history effects in PDM. Specifically, the direction of bias reversal is always from *small* to *large* as long as the feedback in conjunction with the choice indicates $CL = small$ (e.g., $\{S = 0 \rightarrow -1 \rightarrow -2, C = small, F = correct\}$ or $\{S = 0 \rightarrow -1 \rightarrow -2, C = large, F = incorrect\}$) and always from *large* to *small* as long as the feedback in conjunction with the choice indicates $CL = large$ (e.g., $\{S = 0 \rightarrow 1 \rightarrow 2, C = large, F = correct\}$ or $\{S = 0 \rightarrow 1 \rightarrow 2, C = small, F = incorrect\}$). Critically, the value-updating scenario does not predict the bias reversal (Fig 3E, right). It predicts that the feedback effects only asymptotically decrease as a function of sensory evidence but never switch to the other direction. This is because the decision confidence, $p(CL = large(small))$, only modulates the amount of value-updating but never changes the direction of value-updating.

## Ex ante simulation of the feedback effects under the 2 scenarios

Above, we have conceptually explained why and how the 2 scenarios imply the distinct patterns of stimulus-dependent feedback effects. Though this implication seems intuitively apparent, it must be confirmed under the experimental setting of the current study. Moreover, there are good reasons to expect any history effect to exhibit complex dynamics over trials. First, sensory and mnemonic measurements are subject to stochastic noises, which propagates through decision-making and value/boundary-updating processes to subsequent trials (e.g., a sensory measurement that happens to fall on a relatively *small* side is likely to lead to a *small* choice, which affects the subsequent value/boundary-updating process, and so on). Second, provided that any deterministic value/boundary-updating processes are presumed to be at work, the PDM episode on a given trial must, in principle, be probabilistically conditioned on the episodes in past trials (e.g., the current *small* choice on the ring of $S = 0$ is likely to have followed the previous episodes leading to "boundary-updating in the *large* direction" or "positive value-updating of the *small* choice"). Third, 2 steps of deterministic value/boundary-updating occur between what can be observed at $toi-1$ and at $toi+1$ (as indicated by the psychometric curves in Fig 4A), once following the episode at $toi-1$ ($U_{toi-1}$ in Fig 4A) and next following the episode at $toi$ ($U_{toi}$ in Fig 4A). Thus, the differences between the retrospective and prospective history effects should be construed as reflecting not only $U_{toi}$ but also $U_{toi-1}$. The nuanced impacts of this hidden updating on the history effects must be complicated and thus be inspected with realistic simulations. Further, considering that these multiple stochastic and deterministic events interplay to create diverse temporal contexts, history effects are supposed to reveal themselves in multiplexed dynamics.

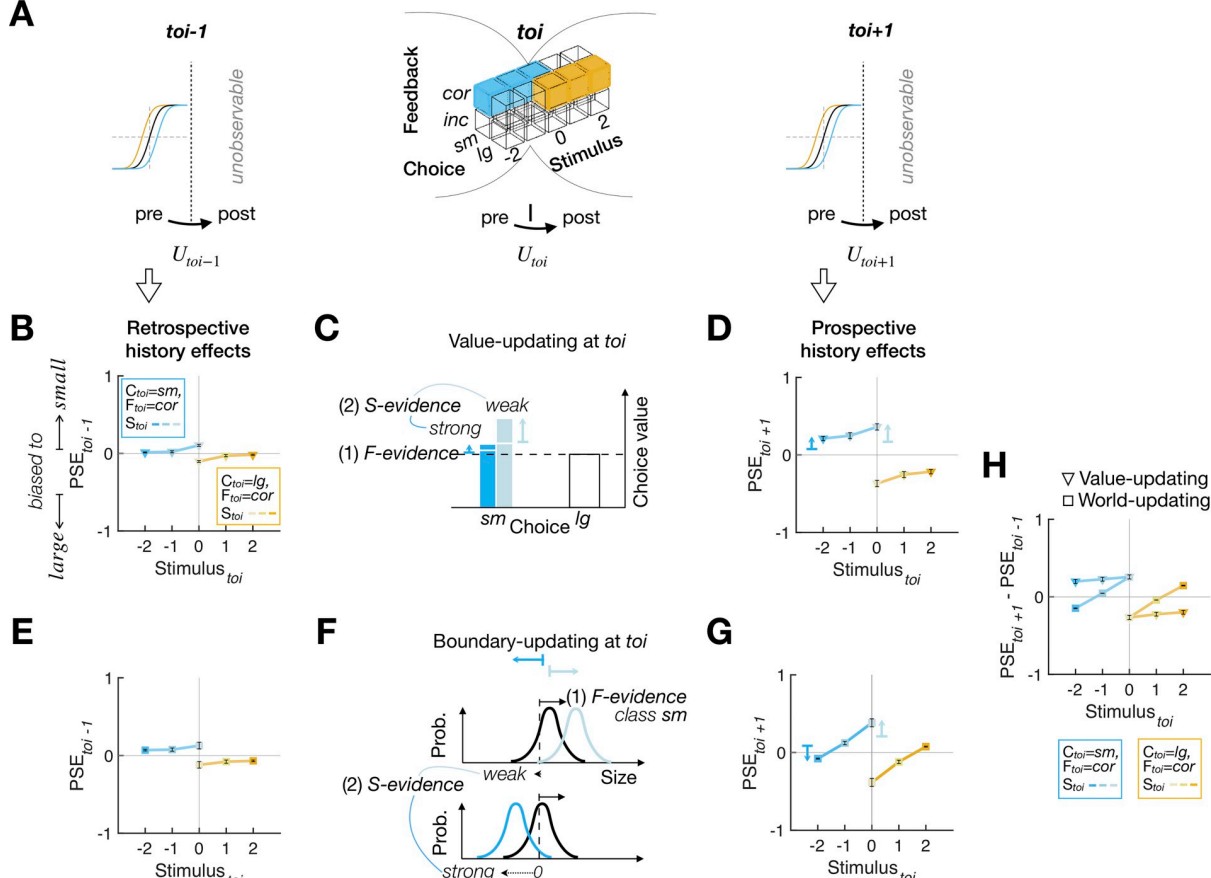

**Fig 4. Ex ante simulation results for the PDM episodes with *correct* feedback. (A)** Illustration of how the retrospective (left) and prospective (right) history effects relate to the value updates and boundary updates (bottom) occurring over the trials overarching the trial of interest. While the updating occurs latently at every trial (as indicated by $U_{toi-1}$, $U_{toi}$, $U_{toi+1}$), its behavioral consequences are observable only at the pre-updating phase at *toi*−1 and *toi*+1. **(B-D)** The observable retrospective **(B)** and prospective **(D)** history effects and latent value-updating processes **(C)** for the value-updating model agent. **(C)** Since *correct* feedback is treated as a positive reward, the chosen value is updated positively while the amount of value-updating varies depending on the strength of sensory evidence, as indicated by the length of the vertical arrows in different colors (weak sensory evidence, pale blue; strong sensory evidence, dark blue). The short horizontal bars and arrow heads of the colored arrows indicate the chosen values before and after $U_{toi}$, respectively. **(E-G)** The observable retrospective **(E)** and prospective **(G)** history effects and latent boundary-updating processes **(F)** for the world-updating model agent. **(F)** Since *correct* feedback is treated as a logical indicator of the true state of the class variable (i.e., the true inequality between the class boundary and the stimulus), the class boundary shifts as a joint function of feedback and sensory evidence, where the boundary shift due to sensory evidence (solid black arrows) counteracts that due to feedback (dotted black arrows), as indicated by the arrows in different colors (weak sensory evidence, pale blue; strong sensory evidence, dark blue). The short vertical bars and arrow heads of the colored arrows at the top indicate the class boundary before and after $U_{toi}$, respectively. **(H)** Juxtaposition of the differences between the retrospective and prospective history effects displayed by the 2 model agents. **(C, F)** The contributions of both sensory and feedback evidence are indicated by *S-evidence* and *F-evidence*, respectively. **(B, D, E, G)** Data points are the means and SEMs across the parameter sets used in ex ante simulations (see Materials and methods). The data underlying this figure **(B, D, E, G, H)** can be found in S1 Data.

Hence, we simulated ex ante the 2 models over a reasonable range of parameters by making the model agents perform the binary classification task on the sequences of stimuli that will be used in the actual experiment (Table A in S1 Appendix, S4 Fig, and Materials and methods). The simulation results confirmed our intuition, as summarized in Fig 4, which shows the retrospective and prospective history effects for the PDM episodes with *correct* feedback. Notably, the retrospective history effects indicate that both value-updating and world-updating agents were already slightly biased to the choice they are about to make in the—following—*toi* (Fig 4B and 4E). One readily intuits that such retrospective biases are more pronounced when

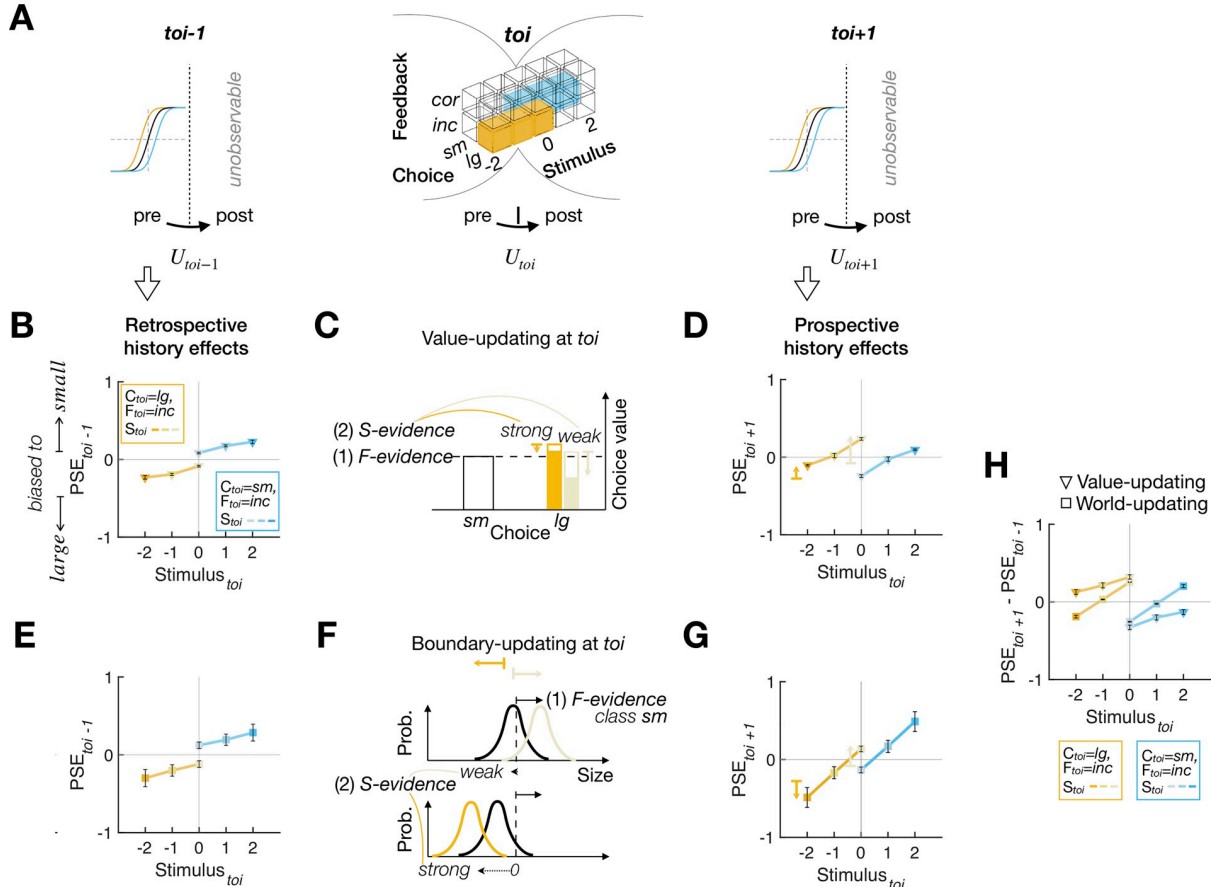

**Fig 5. Ex ante simulation results for the PDM episodes with *incorrect* feedback.** The format is identical to that in Fig 4. The data underlying this figure **(B, D, E, G, H)** can be found in S1 Data.

conditioned on the *toi* with weak sensory evidence because the stochastic bias consistent with the choice that would be made in the *toi* is required more in those trials. This testifies to the presence of the complex dynamics of history effects discussed above and is also consistent with what has been previously observed (e.g., see Fig 2 of the previous study [9]). Importantly, in line with our conceptual conjecture (Fig 3D and 3E), the 2 agents evidently disagree on the prospective history effects. While the value-updating agent always exhibits the feedback-congruent bias but never reverses the direction of bias, the world-updating agent shows the feedback-congruent bias after viewing the ambiguous stimulus but progressively reversed the direction of bias as the stimulus evidence supporting the decision becomes stronger (Fig 4C, 4D and 4F–4H).

Next, Fig 5 summarizes the history effects for the PDM episodes with *incorrect* feedback. The retrospective history effects show that both agents exhibit the choice bias consistent with the choice they will make next trial, as in the case for *correct* feedback, but the amounts of bias are much greater compared to those in the *correct*-feedback condition (Fig 5B and 5E). These pronounced retrospective effects conditioned on the *incorrect*-feedback episodes are intuitively understood as follows: The value-updating agent's value ratio or the world-updating agent's class boundary was likely to be somehow "unusually and strongly" biased before the *toi*, given that they make an *incorrect*—thus "unusual"—choice in the *toi*. Supporting this intuition, the

retrospective bias increases as sensory evidence increases, since the prior value ratio or class boundary must be strongly biased to result in that particular *incorrect* choice despite such strong sensory evidence. Importantly, despite these large retrospective biases, the prospective history effects indicate that both agents adjust their value and class boundary, respectively, in their own manners identical to those for the *correct*-feedback episodes (Fig 5C, 5D, 5F and 5G). Thus, as in the case of the *correct*-feedback episodes, the direction reversal is displayed only by the world-updating agent, but not by the value-updating agent (Fig 5H).

In sum, the ex ante simulation confirmed that the bias reversal of the stimulus-dependent feedback effects occurs only under the world-updating scenario but not under the value-updating scenario, regardless of the (*correct* or *incorrect*) states of feedback. The simulation results also confirmed that, with the current experimental setting, we can empirically determine which of the 2 scenarios provides a better account of feedback effects.

## Evaluating the 2 scenarios for the goodness of fit to human decision-making data

Having confirmed the distinct predictions of the 2 scenarios via ex ante simulation, we evaluated their goodness of fit to human data. As points of reference for evaluation in the model space (Fig 6A), we created 3 reference models. The "Base" model sets the class boundary at the

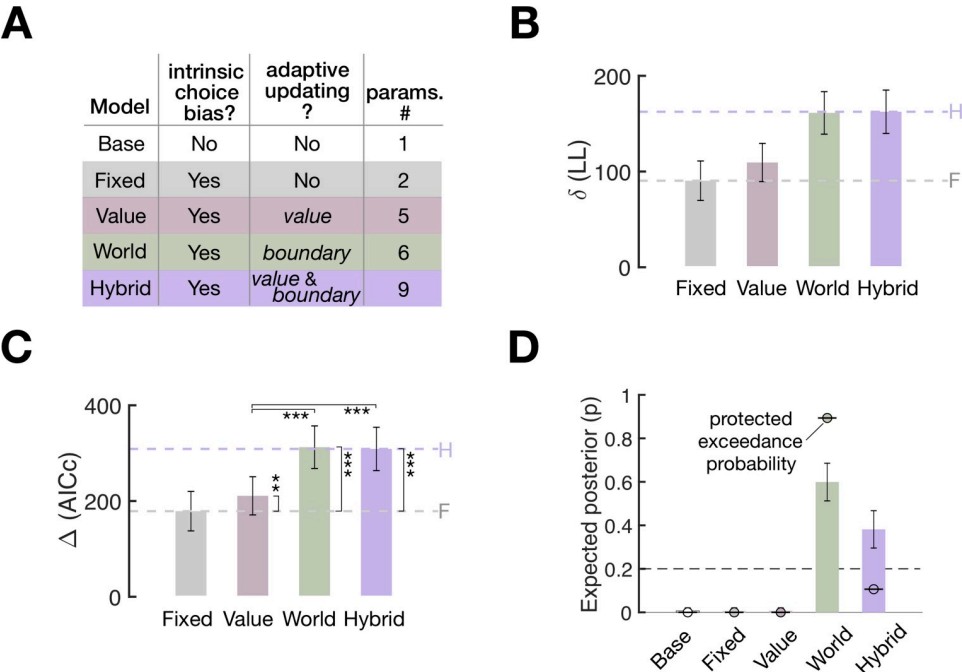

**Fig 6. Model goodness of fit to human choice behavior.** (A) Specification of the models constituting the model space. The color labels also apply to the rest of the panels in (B-D). (B, C) Model comparisons in goodness of fit in terms of log likelihood (B) and AICc (C). The height of bars represents the across-participant average differences from the goodness of fit measures of the Base model ($N = 30$, mean ± SEM). Both difference measures indicate a better fit for higher values. Dashed lines in purple (Hybrid model) and gray (Fixed model) provide the reference points for evaluating the value-updating and world-updating models' accountability of the trial-to-trial choice variability (see main text for their exact meanings). Pairwise model comparisons were performed using paired one-tailed $t$ tests (asterisks indicate significance: *, $P < 0.05$; **, $P < 0.005$; ***, $P < 10^{-8}$) (D) Model comparisons in the hierarchical Bayesian model selection measures. Height of bars, expected posterior probabilities; error bars, standard deviation of posterior probabilities. Dots marked with short dashes, protected exceedance probability. Dashed lines, chance level ($p = 0.2$), indicating the probability that a model is favored over others in describing the data by random chance. Bayesian omnibus risk (BOR), the estimated probability that observed differences in model frequencies may be due to chance, is reported (BOR = $1.7636 \times 10^{-10}$). The data underlying this figure (**B, C, D**) can be found in S1 Data.

unbiased value ($B = 0$) and does not update any choice values, thus incorporating neither arbitrary choice preference nor adaptive updating. The "Fixed" model is identical to the Base model except that it incorporates arbitrary choice preference by fitting the constant class boundary to the data. The "Hybrid" model incorporated both value-updating and world-updating algorithms. We quantified the models' ability to predict human classification choices using log likelihood (Fig 6B) and compared their abilities using the Akaike information criterion corrected for sample size (AICc [32]; Fig 6C)).

The Fixed model's performance relative to the Base model's (gray dashed lines in Fig 6B and 6C) reflects the fraction of choice variability that is attributed to arbitrary choice preference. On the other hand, the Hybrid model's performance relative to the Base model's (purple dashed lines in Fig 6B and 6C) reflects the maximum fraction of choice variability that can be potentially explained by either the value-updating model, the world-updating model, or both. Thus, the difference in performance between the Hybrid and Fixed models (the space spanned between the gray and purple dashed lines in Fig 6B and 6C) quantifies the meaningful fraction of choice variability that the 2 competing models of interest are expected to capture. Prior to model evaluation, we confirmed that the 2 competing models (the value-updating and world-updating models) and 2 reference models (the Base and Hybrid models) are empirically distinguishable by carrying out a model recovery test (S3 Fig).

With this target fraction of choice variability to be explained, we evaluated the 2 competing models by comparing them against the Fixed and Hybrid models' performances while taking into account model complexity with AICc. The value-updating model was moderately better than the Fixed model (paired one-tailed $t$ test, $t(29) = -2.8540$, $P = 0.0039$) and substantially worse than the Hybrid model (paired one-tailed $t$ test, $t(29) = 7.6996$, $P = 8.6170 \times 10^{-9}$) and the world-updating model (paired one-tailed $t$ test, $t(29) = 8.3201$, $P = 1.7943 \times 10^{-9}$). By contrast, the world-updating model was substantially better than the Fixed model (paired one-tailed $t$ test, $t(29) = -10.3069$, $P = 1.6547 \times 10^{-11}$) but not significantly better than the Hybrid model (paired one-tailed $t$ test, $t(29) = -1.0742$, $P = 0.1458$). These results indicate (i) that the world-updating model is better than the value-updating model in accounting for the choice variability and (ii) that adding the value-updating algorithm to the world-updating algorithm does not improve the accountability of the choice variability.

To complement the above pairwise comparisons, we took the hierarchical Bayesian model selection approach [33–35] using AICc model evidence, to assess how probable it is that each of the 5 models prevails in the population (expected posterior probability; vertical bars in Fig 6D) and how likely it is that any given model is more frequent than the other models (protected exceedance probability; dots with horizontal bars in Fig 6D). Both measures corroborated the outcomes of the pairwise comparisons: The world-updating model predominated in expected posterior probability (0.5992) and protected exceedance probability (0.8938).

In sum, the world-updating scenario was superior to the value-updating scenario in predicting the choice behavior of human participants performing the binary classification task.

## Ex post simulation of the feedback effects under the 2 scenarios

The goodness of fit results summarized above simply indicate that the world-updating model is better than the value-updating model in predicting the trial-to-trial variability in choice behavior while taking into account model complexity. Our study aims to examine whether these 2 competing models of interest can account for the stimulus-dependent feedback effects observed in human decision-makers. To do so, we carried out ex post simulations based on the goodness of fit results [36] by testing whether the value-updating and world-updating models can reproduce the observed stimulus-dependent feedback effects.

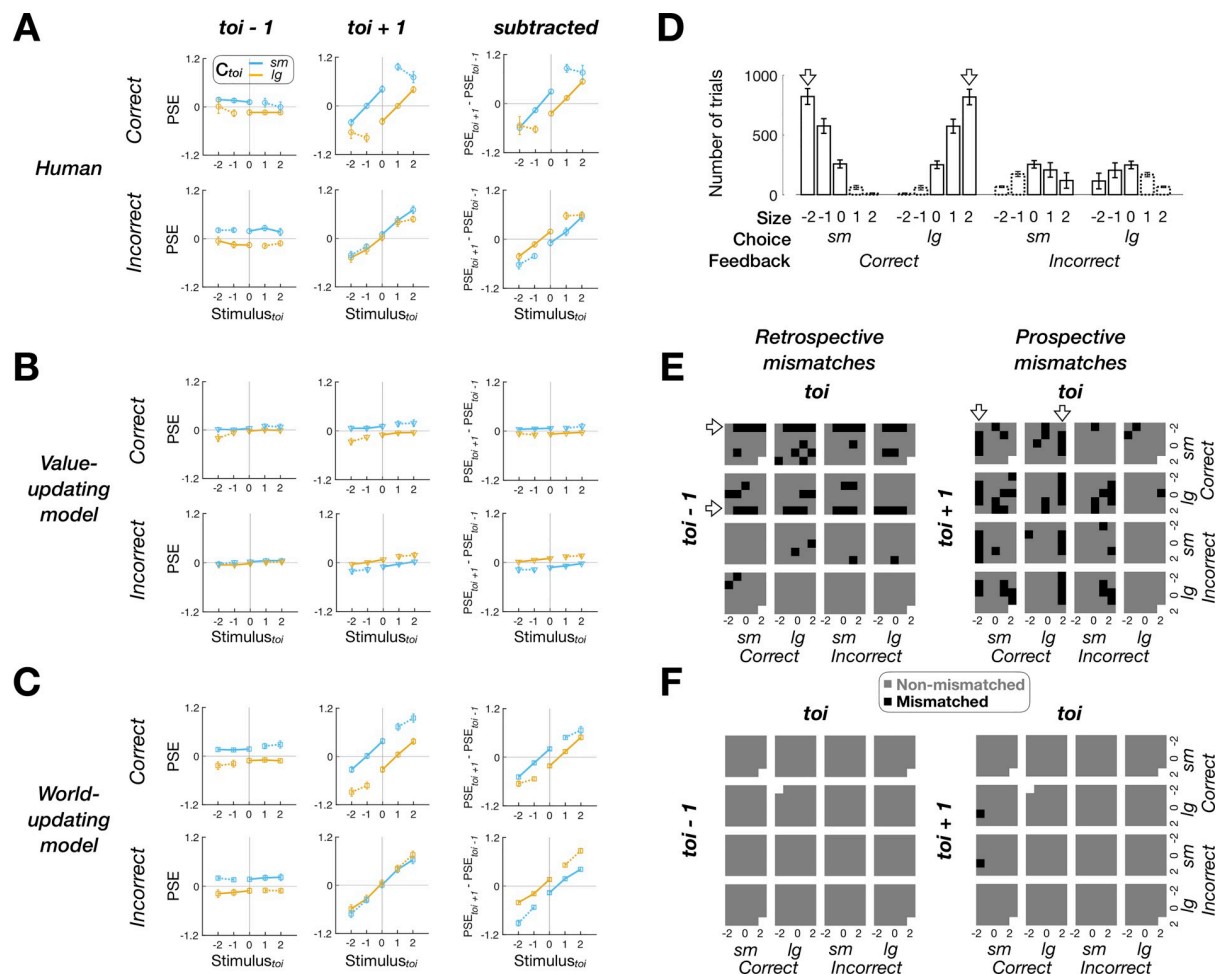

**Fig 7. Ex post simulation results. (A-C)** Retrospective (left columns), prospective (middle columns), and subtractive (right columns) history effects in PSE for the human **(A)**, value-updating **(B)**, and world-updating **(C)** decision-makers. Top and bottom rows in each panel show the PSEs associated with the *toi* episodes involving *correct* and *incorrect* feedback. Symbols with error bars, mean ± SEM across 30 decision-makers. See S5 Fig for the results from the Hybrid model decision-makers. **(D)** Frequency of PDM episodes in the human data (mean and SD across participants). **(E, F)** Maps of significant deviations of the value-updating **(E)** and world-updating **(F)** model agents from the human decision-makers in the retrospective (left) and prospective (right) history effects. Gray and black cells of the maps mark the insignificant and significant deviations (paired two-tailed *t* tests with the Bonferroni correction for multiple comparisons). Empty cells are data points with NaN values due to insufficient trials. The data underlying this figure **(A, B, C, D, E, F)** can be found in S1 Data.

The ex post simulation was identical to the ex ante simulation except that each decision-maker's best-fit model parameters were used (Table B in S1 Appendix; Materials and methods). We assessed how well the models reproduce the human history effects of feedback in 2 different ways. First, we compared the models and the humans similarly to the ex ante simulation (Fig 7A–7C). We included the PDM episodes with nonveridical feedback (symbols with dotted lines in Fig 7A–7C), though those episodes infrequently occurred (12.09 ± 0.02% (mean ± SEM) out of total *toi* episode trials; bars with dotted outlines in Fig 7D). As a result, we inspected the retrospective and prospective history effects, and their differences, for all the possible combinations of "stimulus," "choice," and "feedback" (20 PDM episodes in total), which resulted in a total of 60 PSE pairs to compare. The PSEs simulated by the world-update model closely matched the human PSEs, in both pattern and magnitude (Fig 7A and 7C), whereas those by the value-update model substantively deviated from the human PSEs (Fig 7A and 7B). The statistical comparison (paired two-tailed *t* tests with Bonferroni correction)

indicates that the value-updating model's PSEs significantly deviated from the corresponding human PSEs for almost half of the entire pairs (29 out of 60 pairs), whereas none of the world-updating model's PSEs significantly differed from the human PSEs (0 out of 60 pairs). Notably, most mismatches occurred because the value-updating model does not reverse the direction of feedback effects as sensory evidence becomes stronger while humans do so (compare the third columns of Fig 7A and 7B).

Second, we compared the models and the humans in the probability distribution of retrospective and prospective episodes conditioned on each episode of *toi* (Fig 7D–7F). This comparison allows us to assess the models' reproducibility not just for feedback effects but also for the history effects in general and to explore the origin of the value-based model's failure. By collapsing all the preceding and following trials onto each of the 20 *toi* episodes (the columns of Fig 7E and 7F) and computing their probability distributions across—again—the 20 types of *toi*−1 and 20 types of *toi*+1 episodes (the rows of Fig 7E and 7F), respectively, we could create 400 joint-probability cells.

We carried out repeated *t* tests with Bonferroni correction to see where the model-human mismatches occur (data were missing for a few cells—mostly those including nonveridical-feedback episodes, as indicated by the empty cells in Fig 7E and 7F, because those episodes were too rare (Fig 7D) to occur for all participants). For the remaining cells, the world-updating model showed a remarkable level of correspondence with the humans, deviating from the humans at only 2 cells (out of 790 cells, 0.25%; Fig 7F). By contrast, the value-updating model failed to match the humans for 94 cells (out of 792 cells, 11.87%; Fig 7E). Here, the mismatches occurred systematically: They were frequent when the preceding episode defining any given cell (i.e., episodes at *toi*−1 for the retrospective cells or episodes at *toi* for the prospective cells) was featured with strong sensory evidence (as indicated by the arrows in Fig 7E). This systematic deviation precisely reflects the incapability of the value-updating model to reverse the direction of feedback effects as sensory evidence strengthens.

In sum, the stimulus-dependent history effects of feedback observed in humans could be reproduced by the world-updating scenario but not by the value-based scenario.

## Discussion

Here, we explored the 2 possible scenarios for what humans learn from corrective feedback in a PDM task. We implemented the value-updating scenario with the belief-based RL model [9,10], originally developed to account for the stimulus-dependent effects of reward feedback on animals' PDM. As an alternative, we implemented the world-updating scenario with BMBU, where decision-makers continuously update their internal knowledge about stimulus distribution based on sensory measurements and corrective feedback. The latter excels over the former in predicting the choice behavior and reproducing the stimulus-dependent feedback effects in human PDM, suggesting that humans update their knowledge about world statistics upon corrective feedback for PDM.

Given RL models' success in VDM and the presence of physical rewards, it is not surprising for the belief-based RL model to be considered as an account of the feedback effects in animals' PDM. The original work [9] supported this model using 6 datasets, including 1 human dataset [37]. However, the current work indicates that the way humans learn from corrective feedback —without any physical or monetary reward—in PDM deviates from the value-updating scenario. The critical deviation occurred for the PDM episodes with strong sensory evidence: Past *correct* feedback should, albeit weakly, reinforce the choice made in the past according to the value-updating scenario, whereas humans made the opposite choice more frequently. In fact, the human dataset previously analyzed in the study [9] exhibits the same deviations (see their

Fig 8C and 8D). When this dataset was analyzed in our way, it displayed the patterns almost identical to those of our dataset (S7A Fig). For that matter, another published human dataset [31] substantially deviated from the value-updating scenario (S7B Fig). We remain cautious about the possibility that even animals may demonstrate such deviations as well. However, this possibility seems worth exploring though, given that the main dataset from the 16 rats engaged in an olfactory PDM task also exhibited patterns similar to those found in humans when corrected for the bias present in previous trials (see Fig 2i in the study [9]). Notably, in these studies [9,31,37], the class boundary existed either implicitly (e.g., a perfectly balanced odor mixture [9]) or explicitly (e.g., a reference stimulus presented in another interval [37]). This suggests the possibility that the bias reversal of feedback effects may be a general phenomenon that can be observed in diverse types of binary classification tasks. However, further empirical tests are required to confirm this possibility. The bias reversal of feedback effects should not be treated lightly as a nuisance because any variant of the RL algorithm cannot reverse the direction of reinforcement in principle, as demonstrated in our work and in the modeling results of the same study [9] (shown in their Fig 3). By contrast, BMBU provides a principled account of these effects by treating *correct* and *incorrect* feedback as what they supposedly mean, a teaching signal indicating the true state of the class variable.

To be sure, the idea of shifting the decision or class boundary toward past stimuli per se is not new and has been previously hypothesized [38,39] or implemented into various models [40–44]. However, BMBU goes beyond these efforts by offering a normative formalism of incorporating *correct* and *incorrect* feedback as evidence for the class boundary such that it has an equal footing as sensory evidence in PDM tasks. This integration of feedback and sensory evidence within the framework of BDT advances the current computational account of the history effects because it addresses the history factors in the complete dimensions of PDM ("stimulus," "choice," and "feedback"), which is important given the multiplexed nature of history effects emphasized by prior studies [8–11,31,45]. Our modeling work joins recent computational and empirical efforts of incorporating feedback in the normative evidence accumulation model [6,46], a framework commonly employed in various classic PDM tasks, such as a random-dot motion task. Furthermore, a study on rats' binary classification behavior has shown that rats can use information about the correct class state (referred to as "second-order prior" by the authors) by integrating their own choices with feedback (reward outcome) and that the population neural activity in the orbitofrontal cortex represents this information [11]. Together with these studies, our work supports a general view that decision-makers use corrective feedback as evidence for updating their world knowledge pertinent to the PDM task engaging them. Having mentioned the general view on the role of feedback in human PDM, future efforts are needed to further verify the stimulus-dependent feedback effects under various sensory modalities and PDM tasks.

Previously, the so-called "Anna Karenina" account was presented to describe the seemingly idiosyncratic *incorrect* feedback effects [9]. The Anna Karenina account leaves the crucial aspect of feedback effects—the different consequences of *correct* versus *incorrect* feedback— unexplained. Since the belief-based RL model predicts the specific pattern of feedback effects for incorrect trials, as shown via ex ante simulation, endorsing the Anna Karenina account admits that the belief-based RL model fails to account for the effects of *incorrect* feedback observed in animals. For that matter, past studies on the history effects in PDM paid little attention to incorrect trials because they are, owing to their infrequency, considered too noisy and unreliable to be properly analyzed. By contrast, BMBU accounts for the effects of feedback in a principled way, regardless of whether the feedback is *correct* or *incorrect*. Furthermore, BMBU explains why the feedback effects appear different between the correct and incorrect trials on the surface (compare the prospective history effects between Figs 4 and 5): The correct

and incorrect trials share the same deterministic boundary-updating process but had different histories of their own stochastic events, which led to correct versus incorrect choices, respectively.

As mentioned earlier, the history effects are dynamic and multiplexed in nature. This calls for an effort to establish a rigorous framework to probe behavioral data for the history effects. Several recent studies made such efforts by taking various approaches, yet all emphasizing the presence of distinct sources of biases. One study [47] assumed 2 sources with differing time scales and took a regression-based approach to separate their influences on choice bias by incorporating them as independent regressors to predict choices. Another group of researchers [6,9] also noted the presence of slow fluctuations and raised a concern about the conventional practice of inspecting only the prospective history effects because nonsystematic slow fluctuations in the decision-making strategy may cause the observed effects. This group dealt with this concern by subtracting the retrospective history effects from the prospective ones. A more recent study [48] shared this concern but disagreed about its remedy by showing that the subtraction method cannot fairly recover diverse systematic updating strategies. Alternatively, they took a model-based approach to separate any given updating strategy from random drifts in decision criteria. We acknowledge the importance of the efforts by these studies and share the same concern. But, we emphasize that BMBU successfully reproduced human history effects in both directions of time without incorporating any nonsystematic components arising from random drifts. BMBU's concurrent reproduction of the retrospective and prospective history effects was confirmed not just for the summary statistics (the PSEs in Fig 7C) but also for the individual data points spanning almost the entire space of PDM episode pairs (Fig 7F). This suggests that it is an empirical matter of whether the decision criterion slowly drifts or not, raising another concern that systematic history effects might be explained away as nonexisting slow drifts. In this sense, we propose that researchers should treat the retrospective history effects not as a baseline or control condition but as what must be explained, the phenomenon equally important as the prospective history effects, before resorting to any nonsystematic sources. We believe that such a treatment is the way historians treat historical events [49] and that our approach showcases its one rigorous example.

## Materials and methods

### Ethics statement

The study protocol was approved by the Seoul National University Institutional Review Board (No. 1310/001-020). All the experiments were conducted in accordance with the principles expressed in the Declaration of Helsinki. All participants gave prior written informed consent to participate in the experiments.

### Participants

All participants (13 females and 17 males, aged 18 to 30 years) were recruited from the Seoul National University (SNU) community and were compensated approximately $10/h.

### Procedure

**Stimuli.** The stimulus was a thin (.07 degree in visual angle (DVA)), Gaussian-noise filtered, black-and-white ring flickering at 20 Hz on a gray luminance background. On each trial, a fixation first appeared for 0.5 s on average (fixation duration uniformly jittered from 0.3 s to 0.7 s on a trial-to-trial basis) before the onset of a ring stimulus. Five different ring sizes

(radii of 3.84, 3.92, 4.00, 4.08, 4.16 DVA, denoted by −2, −1, 0, 1, 2, respectively, in the main text) were randomized within every block of 5 trials.

**Task.** Participants performed a binary classification task on ring size with trial-to-trial corrective feedback. Each individual participated in 5 daily sessions, each consisting of 6 runs, each consisting of 170 trials, ended up performing a total of 5,100 trials. In any given trial, participants viewed one of the 5 rings and indicated its class (*small* or *large*) within 1.2 s after stimulus onset by pressing one of the 2 keys using their index and middle fingers. The assignment of computer keys for *small* and *large* choices alternated between successive sessions to prevent any unwanted choice bias possibly associated with finger preference. The response period was followed by a feedback period of 0.5 s, during which the color of the fixation mark informed the participants of whether their response was correct (green) or not (red). In case no response had been made within the response period, the fixation mark turned yellow, reminding participants that a response must be made in time. These late-response trials comprised 0.5418% of the entire trials across participants and were included in data analysis. Meanwhile, the trials on which a response was not made at all comprised 0.0948% of the entire trials. These trials were excluded from analysis and model fitting. As a result, the number of valid trials per participant ranged from 5,073 to 5,100 with an average of 5,095.2 trials. Before each run, we showed participants the ring stimulus of the median size (4.00 DVA in radius) on the screen for 15 s while instructing them to use that ring as a reference for future trials, i.e., to judge whether a test ring is smaller or larger than this reference ring. This procedure was introduced for the purpose of minimizing any possible carryovers from the belief they formed about the class boundary in the previous session. Participants were encouraged to maximize the fraction of correct trials.

**Feedback manipulation.** We provided participants with stochastic feedback using a "virtual" criterion sampled from a normal distribution $N(\mu_{True}, \sigma_{True})$. $\sigma_{True}$ was always fixed at 1.28 throughout the entire runs. In each run, $\mu_{True}$ was initially (up to 40 to 50 trials) set to 0 and then to one of the 3 values ($\mu_{True} = \{-0.4, 0, 0.4\}$) with the equal proportion (10 runs for each value) for the rest of trials. The stochastic feedback was introduced this particular way to create PDM episodes with (occasional) nonveridical feedback while mimicking a real-world situation where references are slightly noisy and biased in an unnoticeable manner.

## Data analysis

For any given PDM episode at a *toi*, we quantified the retrospective and prospective history effects by probing the psychometric curves at the trials before and after *toi*, respectively. The psychometric function ($\psi(x)$) was estimated by fitting the cumulative Gaussian distribution ($F$) to the curves using *Psignifit* package [50–52] (https://github.com/wichmann-lab/psignifit), as follows:

$$\psi(x; \mu, \sigma) = F(x; \mu, \sigma),$$

where $\mu$ and $\sigma$ are the mean and standard deviation of $F$. By finding the best-fitting value of $\mu$, we defined the PSE (the stimulus level with equal probability for a *small* or *large* choice), which was used as the summary statistics that quantifies the history effects associated with a given PDM episode. To ensure reliable PSE estimates, we acquired bootstrap samples ($N = 5,000$) of psychometric curves based on the binomial random process and took their average as the final estimate for each PDM episode. In our main data analysis, the results of which are displayed in Fig 7, we chose not to include the parameters for guess or lapse rates in estimating PSEs. This was done to prevent unfair overfitting problems from occurring in infrequent episode types with small numbers of trials available for fitting. On the other hand, to preclude any potential confounding problem related to the task difficulty associated with PDM

episode types, we also repeated the above PSE estimation procedure with guess ($\gamma$) and lapse ($\lambda$) rates included as free parameters: $\psi(x; \mu, \sigma, \gamma, \lambda) = \gamma + (1 - \gamma - \lambda)F(x; \mu, \sigma)$. The results did not differ between the original estimation procedure without the lapse and guess rates and the procedure with the lapse and guess rates (Bonferroni-corrected $P = 0.2023 \sim 1.000$; paired two-tailed $t$ tests; see S2 Data for detailed statistical information).

## Value-updating model

As a model of the value-updating scenario, we used the belief-based RL model proposed in the previous work [9,10]. This model incorporates RL algorithm into the conventional Bayesian formalism of decision confidence—also known as statistical decision confidence using a partially observable Markov decision process (Fig 3E). In this model, the decision-maker, given sensory measurement $m$, computes the probability that the stimulus belongs to "*large*" ($p_L$) or "*small*" ($p_S = 1 - p_L$) class (hereinafter the $p$-computation), where $p_L = \int_{\mu_0}^{\infty} p(S|m)dS$. This probability will be referred to as a "belief-state," as in the original work [9,10]. Here, the probability distribution $p(S|m)$ is defined as a normal distribution with mean $m$ and standard deviation $\sigma_m$. Whereas $\mu_0$ was assumed to be zero in the original work, we set $\mu_0$ free as a constant parameter to allow the belief-based RL model to deal with any potential individuals' idiosyncratic choice bias, as we will allow the world-updating model (BMBU) to do so (see below). Next, the expected values of the 2 choices $Q_S$ and $Q_L$ can be obtained by $p_S$ and $p_L$ multiplied with the learned values of the options of *small* and *large*, $V_S$ and $V_L$, respectively. Accordingly, the expected value $Q_C$ is also defined separately for the choice made between *small* and *large*: $Q_S$ and $Q_L$.

In the original work, the argmax rule was applied to determine the choice (i.e., the higher $Q$ determines the choice $C$). Instead, here, we applied the softmax rule, which selects *large* with probability $\frac{\exp(\beta Q_L)}{\exp(\beta Q_S) + \exp(\beta Q_L)}$ (the higher $Q$ preferentially selects $C$) where $\beta$ is an inverse temperature. This feature did not exist in the original model but was introduced here to allow the belief-based RL model to deal with stochastic noise at the decision stage, as we allow the world-updating model (BMBU) to do so.

The initial values of *small* and *large* choices were set identically as a free parameter $V_{init}$. Upon receiving feedback on the decision, the decision-maker updates the value of the selected choice $V_C$ by the reward prediction error $\delta$ with learning rate $\alpha$:

$$V_C \leftarrow V_C + \alpha\delta. \tag{1}$$

No temporal discounting is assumed for simplicity. Since the decision-maker treats corrective feedback as rewards (*correct*: $r = +1$, *incorrect*: $r = 0$), the reward prediction error $\delta$ is computed as the deviation of the reward from the expected value:

$$\delta = r - Q_C = r - p_C V_C. \tag{2}$$

Note that the belief state $p_C$ (i.e., statistical decision confidence) modulates $\delta$ such that $\delta$ increases as $p_C$ decreases, which is the crucial relationship constraining the belief-based RL model's key prediction on the stimulus-dependent feedback effects. Specifically, upon *correct* feedback, $\delta$ will take a positive value and reinforce the choice value. However, as $p_C$ increases, the magnitude of such reinforcement will decrease. Critically, despite the decrease of reinforcement as a function of $p_C$, the sign of reinforcement will never be reversed until the expected value $Q$ reaches the maximum reward value ($r = 1$). Based on the same ground, the sign of reinforcement will never be reversed either in the case of *incorrect* feedback. The free parameters of the value-updating model are $\theta = \{\mu_0, \sigma_m, \alpha, \beta, V_{init}\}$.

## World-updating model

As a model of the world-updating scenario, we developed the BMBU. BMBU shares the same platform for PDM with the belief-based RL model (as depicted in Figs 1A and 3A) but, as a BDT model, makes decisions using its "learned" generative model while continually updating its belief about the class boundary $B$, the key latent variable of that internal model (as depicted in the left panel of Fig 3D).

**"Learned" generative model.** In BDT, the learned generative model refers to the decision-maker's subjective internal model that relates task-relevant variables ($m$, $m'$, and $B$ in the left panel of Fig 3D) to external stimuli and behavioral choices ($S$ and $CL$, respectively, in the left panel of Fig 3D). As previously known [53,54], the decision-maker's internal model is likely to deviate from the "actual" generative model that accurately reflects how the experimenter generated external stimuli due to one's limitations in the sensory and memory apparatus. In the current experimental setup, we assumed that the internal model of the decision-maker deviates from that of the experimenter in the following aspect: Due to the noise in the sensory and memory encoding processes, the decision-maker is likely to believe that many rings of different sizes are presented, although the experimenter used only 5 discrete-size rings. The post-experiment interviews supported this: None of the participants reported perceiving discrete stimuli during the experiment. A deviation like this is known to occur commonly in psychophysical experiments where a discrete number of stimuli were used [40,54,55].

We incorporated the above deviation into the decision-maker's internal model by assuming that the stimulus at any given trial is randomly sampled from a Gaussian distribution with mean $B$ and variance $\sigma_S^2$ (as depicted by $B{\rightarrow}S$ in Fig 3D):

$$p(S|B) = N(S; B, \sigma_S^2), \tag{3}$$

which defines the probability distribution of stimuli conditioned on the class boundary, where $\sigma_S^2$ corresponds to the extent to which a given decision-maker assumes that stimuli are distributed. Next, the inequality between the class boundary and the stimulus determines the state of the class $CL$ (as depicted by the converging causal relations involving the class variable, $B{\rightarrow}CL{\leftarrow}S$, in Fig 3D):

$$CL = large(small) \; if \; S > (<)B, \tag{4}$$

which defines the correct answer of the perceptual task. On the other hand, the sensory measurement $m$ at any given trial is randomly sampled from a Gaussian distribution with mean $S$ and variance $\sigma_m^2$ (as depicted by $S{\rightarrow}m$ in Fig 3D):

$$p(m|S) = N(m; S, \sigma_m^2), \tag{5}$$

which defines the probability distribution of sensory measurements conditioned on the stimulus, where $\sigma_m^2$ corresponds to the extent to which the decision-maker's sensory system is noisy. Lastly, the mnemonic measurement $m'$ at any given trial is randomly sampled from a Gaussian distribution with mean $m$ and variance $\sigma_{m'}^2$ (as depicted by $m{\rightarrow}m'$ in Fig 3D):

$$p(m'|m) = N(m'; m, \sigma_{m'}^2), \tag{6}$$

which defines the probability distribution of mnemonic measurements conditioned on the sensory measurement, where $\sigma_{m'}^2$ corresponds to the extent to which the decision-maker's working memory system is noisy. This generative process ($m{\rightarrow}m'$) is required because the sensory evidence of the stimulus is no longer available in the sensory system—due to a brief (0.3 s;

Fig 2B) stimulus duration—at the moment of updating the state of the class boundary (as will be shown below in the subsection titled "Boundary-updating") and instead must be retrieved from the working memory system. The mnemonic recall of the stimulus is known to be noisy, becoming quickly deteriorated right away after stimulus offset, especially for continuous visual evidence such as color and orientation [56,57]. The generative process relating $m$ to $m'$ has been adopted for the same reason by recent studies [58,59], including our group [55], and is consistent with the nonzero levels of memory noise in the model-fit results ($\sigma^2_{m'}$ = [1.567, 5.606]). The substantial across-individual variability of the fitted levels of $\sigma^2_{m'}$ is also consistent with the previous studies [55,58,59].

With the learned generative model defined above, the decision-maker commits to a decision by inferring the current state of the class variable $CL$ from the current sensory measurement $m$ and then updates the current state of the boundary variable from both the current mnemonic measurement $m'$ and the current feedback $F$.

**Decision-making.** As for decision-making, BMBU, unlike the belief-based RL model, does not consider the choice values but completely relies on the $p$-computation by selecting the *large* class if $p_L > 0.5$ and the *small* class if $p_L < 0.5$. The $p$-computation is carried out by propagating the sensory measurement $m$ within its learned generative model:

$$p_L = \int_{\hat{B}}^{\infty} p(S|m)dS, \tag{7}$$

where the finite limit of the integral is defined by the inferred state of the boundary $\hat{B}$, which is continually updated on a trial-to-trial basis (as will be described below). This means that the behavioral choice can vary depending on $\hat{B}$ even for the same value of $m$ (as depicted in the "perception" stage of Fig 3A and 3B).

**Boundary-updating.** After having experienced a PDM episode in any given trial $t$, BMBU (i) computes the likelihood of the class boundary by concurrently propagating the mnemonic measurement $m'_t$ and the "informed" state of the class variable $CL_t$, which can be informed by feedback $F_t$ and choice $C_t$ in the current PDM episode, within its learned generative model ($p(m'_t, CL_t|B_t)$) and then (ii) forms a posterior distribution of the class boundary ($p(B_t|m'_t, CL_t)$) by combining that likelihood with its prior belief about the class boundary at the moment ($p(B_t)$), which is inherited from the posterior distribution formed in the previous trial $t - 1$ ($p(B_{t-1}|m'_{t-1}, CL_{t-1})$). Intuitively put, as BMBU undergoes successive trials, its posterior belief in the previous trial becomes the prior in the current trial, being used as the class boundary for decision-making and then being combined with the likelihood to be updated as the posterior belief in the current trial. Below, we will first describe the computations for (i) and then those for (ii). As explained above (Eq 6), we stress that the likelihood computation must be based not on the sensory measurement $m_t$ but on the mnemonic measurement $m'_t$ because $m_t$ is no longer available at the moment of boundary-updating.

As for the boundary likelihood computation (i), BMBU posits that the decision-maker infers how likely the current PDM episode—i.e., the combination of the mnemonic measurement $m'_t$, the choice $C_t$, and the corrective feedback $F_t$—is generated by hypothetical values of the class boundary ($p(m'_t, C_t, F_t|B_t)$). Since the "true" state of the class variable $CL_t$ is deduced from any given pair of $C_t$ and $F_t$ states in binary classification as follows,

$$CL_t = large \text{ if } C_t = large \text{ and } F_t = correct \text{ or if } C_t = small \text{ and } F_t = incorrect;$$

$$CL_t = small \text{ otherwise},$$

the likelihood can be defined using only $m'_t$ and $CL_t$ : $p(m'_t, C_t, F_t|B_t) \equiv p(m'_t, CL_t|B_t)$. Hence,

the likelihood of the class boundary is computed by propagating $m'_t$ and $CL_t$ inversely over the learned generative model (as defined by Eqs 3–6):

$$p(m'_t, CL_t|B_t) = \int p(m'_t, CL_t, S_t|B_t)dS_t = \int p(m'_t|S)p(CL_t|S_t, B_t)p(S_t|B_t)dS_t, \quad (8)$$

which entails the marginalization over every possible state of $S_t$, a variable unknown to the decision-maker. Here, since the binary states of $CL_t$ ($CL_t \in \{small, large\}$) indicates the inequality between $S_t$ and $B_t$ (Eq 4), $B_t$ is used as the finite limit of the integrals to decompose the original integral into the one marginalized over the range of $S_t$ satisfying $CL_t = small$ and the other marginalized over the range of $S_t$ satisfying $CL_t = large$:

$$\int p(m'_t|S)p(CL_t|S_t, B_t)p(S_t|B_t)dS_t =$$

$$= \int_{-\infty}^{B_t} p(m'_t|S_t)p(CL_t|S_t, B_t)p(S_t|B_t)dS_t + \int_{B_t}^{+\infty} p(m'_t|S_t)p(CL_t|S_t, B_t)p(S_t|B_t)dS_t. \quad (9)$$

Note that the boundary likelihood function is computed based on $CL_t$ informed by feedback. The right-hand side of Eq 9 can further be simplified for the informed state $CL_t$ by replacing the infinite limits with finite values (Equation S5 in Text in S1 Appendix). For the case of $CL_t = large$, $p(CL_t|S_t, B_t)$ in the left and right integral terms on the right-hand side of Eq 9 becomes 0 and 1, respectively, while becoming 1 and 0 for the case of $CL_t = small$ in the ranges of $S_t$ of the corresponding integrals (Equation S3-S6 in Text in S1 Appendix). Hence, we find the likelihood of the class boundary in a reduced form, separately for $CL_t = large$ and $CL_t = small$, as follows:

$$p(m'_t, CL_t = small|B_t) = \int_{-\infty}^{B_t} p(m'_t|S_t)p(S_t|B_t)dS_t;$$

$$p(m'_t, CL_t = large|B_t) = \int_{B_t}^{+\infty} p(m'_t|S_t)p(S_t|B_t)dS_t \quad (10)$$

where $p(m'_t|S_t) = N(m'_t; S_t, \sigma_{m'}^2 + \sigma_m^2)$, according to the "chain" relations defined in the learned generative model ($S \to m \to m'$ in the left panel of Fig 3D; Eqs 5 and 6; see Equation S2 for derivations in Text in S1 Appendix). Eq 10 indicates that BMBU calculates how likely hypothetical boundary states bring about the mnemonic measurement ($B \to S \to m \to m'$) while taking into account the informed state of the class variable ($B \to CL \leftarrow S$), by constraining the possible range of the stimulus states. To help readers intuitively appreciate these respective contributions of the mnemonic measurement and the informed state of the class variable (feedback) to the boundary likelihood, we further elaborated on how Eq 9 is reduced to Eq 10 depending on the informed state of $CL_t$ (see Text in SI Appendix and S1 Fig).

Lastly, we evaluate the integral for $CL_t = small$ in Eq 10 by substituting $p(S_t|B_t) = N(S_t; B_t, \sigma_S^2)$ and $p(m'_t|S_t) = N(m'_t; S_t, \sigma_{m'}^2 + \sigma_m^2)$, from the defined statistical knowledge in the

learned generative model (Eq 3 and Eqs 5 and 6, respectively) and find:

$$p\left(m'_t, CL_t = small|B_t\right)$$

$$= \frac{1}{\sqrt{2\pi\left(\frac{\sigma_M^2\sigma_S^2}{(\sigma_M^2+\sigma_S^2)}\right)}} \int_{-\infty}^{B_t} e^{-\frac{\left(S_t - \frac{B_t\sigma_M^2 + m'_t\sigma_S^2}{\sigma_M^2+\sigma_S^2}\right)^2}{2\frac{\sigma_M^2\sigma_S^2}{(\sigma_M^2+\sigma_S^2)}}} dS_t \times \frac{1}{\sqrt{2\pi(\sigma_M^2+\sigma_S^2)}} e^{-\frac{(B_t-m'_t)^2}{2(\sigma_M^2+\sigma_S^2)}}. \qquad (11)$$

where $\sigma_M^2 = \sigma_{m'}^2 + \sigma_m^2$. For the other state in feedback, we evaluate the integral in the same manner and find:

$$p\left(m'_t, CL_t = large|B_t\right)$$

$$= \frac{1}{\sqrt{2\pi\left(\frac{\sigma_M^2\sigma_S^2}{(\sigma_M^2+\sigma_S^2)}\right)}} \int_{B_t}^{\infty} e^{-\frac{\left(S_t - \frac{B_t\sigma_M^2 + m'_t\sigma_S^2}{\sigma_M^2+\sigma_S^2}\right)^2}{2\frac{\sigma_M^2\sigma_S^2}{(\sigma_M^2+\sigma_S^2)}}} dS_t \times \frac{1}{\sqrt{2\pi(\sigma_M^2+\sigma_S^2)}} e^{-\frac{(B_t-m'_t)^2}{2(\sigma_M^2+\sigma_S^2)}}. \qquad (12)$$

Having calculated the likelihood of $B_t$, we turn to describe (ii) how BMBU combines that likelihood with a prior distribution on trial $t$, which forms a posterior distribution of $B_t$ according to Bayes rule:

$$p(B_t|m'_t, CL_t) \propto p(m'_t, CL_t|B_t)p(B_t). \qquad (13)$$

We assumed that, at the beginning of the current trial $t$, the decision-maker recalls the posterior belief $p(B_{t-1}|m'_{t-1}, CL_{t-1})$ formed (Eq 13) from the previous trial—to use it as the prior of $B_t$—into the current working memory space, and it is thus subject both to decay $\lambda$ and diffusive noise $\sigma_{diffusion}$ during the recall process. As a result, the prior $p(B_t)$ is basically the recalled posterior, defined as the normal distribution $N(\hat{B}_t, \sigma_{B_t}^2)$ as follows:

$$\hat{B}_t = \lambda\hat{B}_{t-1}^{post} + (1-\lambda)\mu_0;$$

$$\sigma_{B_t}^2 = \lambda\sigma_{t-1}^{2\,post} + \sigma_{diffusion}^2, \qquad (14)$$

where $\hat{B}_{t-1}^{post}$ and $\sigma_{t-1}^{2\,post}$ denote mean and variance of the previous trial's posterior distribution.

Note that the decay parameter $\lambda = \frac{\sigma_0^2}{\sigma_0^2 + \sigma_{t-1}^{2\,post}}$ influences the width and location of the belief distribution and that the diffusive noise of $\sigma_{diffusion} > 0$ helps to keep the width of the distribution over multiple trials, thus avoiding sharpening and stopping the updating process [60]. In this way, $\lambda$ and $\sigma_{diffusion}$ allow BMBU to address the idiosyncratic choice bias and noise, as we equip the belief-based RL model to do so with $\mu_0$ and the sofmax rule.

In sum, BMBU posits that human individuals carry out a sequence of binary classification trials with their learned generative model while continually updating their belief about the location of the class boundary in that generative model. BMBU describes these decision-making and boundary-updating processes using a total of 6 parameters ($\theta = \{\mu_0, \sigma_m, \sigma_s, \sigma_0, \sigma_{m'}, \sigma_{diffusion}\}$), which are set free to account for individual differences.

## Reference models

As the references for evaluating the belief-based RL model and BMBU in predicting the variability of human choices, we created 3 reference models. The "Base" model captures the choice variability that can be explained by the $p$-computation with the class boundary fixed at 0 unanimously for all participants and without any value-updating process. Thus, it has only a single free parameter representing the variability of the sensory measurement ($\theta = \{\sigma_m\}$). The "Fixed" model captures the choice variability that can be explained by the $p$-computation with the class boundary set free to a fixed constant $\mu_0$ for each participant and without any value-updating process. Thus, it has 2 free parameters ($\theta = \{\mu_0, \sigma_m\}$). The "Hybrid" model captures the choice variability that can be explained both by the $p$-computation with the inferred class boundary by BMBU and by the value-updating process implemented by the belief-based RL model. Thus, it has 9 free parameters ($\theta = \{\mu_0, \sigma_m, \sigma_s, \sigma_0, \sigma_{m'}, \sigma_{diffusion}, \alpha, \beta, V_{init}\}$). In Fig 6B–6D, the differential goodness of fit measures on the y-axis indicate the subtractions of the performance of the "Base" model from those of the remaining models.

## Model fitting

For each participant, we fitted the models to human choices over N valid trials ($N \leq 170$) of M (= 10) experimental runs under K (= 3) conditions, where invalid trials were the trials in which the participants did not make any response. For any given model, we denote the log likelihood of a set of parameters $\theta$ given the data as follows:

$$LL(\theta, model) = \log p(data|\theta, model) = \sum_{k=1}^{K_{cond}} \sum_{j=1}^{M_{runs}} \sum_{i=1}^{N_{trials}} \log p(C_{i,j,k}|\theta, model),$$

where $C_{i,j,k}$ denotes the participant's choice (*large* or *small*) on the $i$-th trial of the $j$-th run under the $j$-th condition. Computation of this $LL$ is analytically intractable given the stochastic nature of choice determination. So, we used inverse binomial sampling (IBS; [61]), an efficient way of generating unbiased estimates via numerical simulations. The maximum-likelihood estimate of the model parameters was obtained with Bayesian Adaptive Direct Search (BADS) [62], a hybrid Bayesian optimization to find the parameter vector $\theta^*$ that maximizes the log likelihood, which works well with stochastic target functions. To reduce the risk of being stuck at local optima, we repeated 20 independent fittings by setting the starting positions randomly using Latin hypercube sampling (*lhsdesign_modifed.m* by Nassim Khlaled; https://www.mathworks.com/matlabcentral/fileexchange/45793-latin-hypercube) and then picked the fitting with the highest log likelihood. To avoid infinite loops from using IBS, we did not impose individual lapse rates in an arbitrary manner. Instead, we calculated the average of the lapse rate and guess rate from the cumulative Gaussian fit to a given individual's grand mean (based on the entire trials) psychometric curve. With these individual lapse probabilities (mean rate of 0.05, which ranged [0.0051, 0.1714]), trials were randomly designated as lapse trials, in which the choice was randomly determined to be either *small* or *large*.

## Model comparison in goodness of fit

We compared the goodness of fit of the models using AICc based on maximum-likelihood estimation fitting, as follows:

$$AICc = -2 \cdot LL(\theta^*) + 2p + \frac{2p(p+1)}{(NxMxK) - p - 1},$$

where $p$ is the number of parameters of the model and the total number of trials in the dataset is

$N \times M \times K$. Log model evidence was obtained for each participant by multiplying AICc by $-1/2$ [35]. Furthermore, we took a hierarchical Bayesian model selection approach that infers the posterior over model frequencies in the population based on log model evidence values in each participant. To conclude whether a given model is the most likely model above and beyond chance, we also reported protected exceedance probabilities for each model (see Fig 6E and 6F). The random effects model selection at the group level relied on the function *VBA_groupBMC.m* of the Variational Bayesian Analysis toolbox (https://mbb-team.github.io/VBA-toolbox/) [63].

## Model recovery analysis

We performed a model recovery analysis to further validate our model fitting pipeline. In the analysis, we considered the 2 competing models of interest (the world-updating and value-updating models) and the 2 reference models (the Base and Hybrid models). Using the same parameter set, we generated synthetic data for each participant's true stimulus sequences. For the realistic synthetic data, the parameter values were chosen based on the best-fitting parameter estimates from each individual. We generated 30 sets of synthetic data for each model, with 153,000 trials in each set. We then fit all 4 models to each synthetic dataset, resulting in 480 fitting problems. We assessed the models using the AICc-based log model evidence and computed exceedance probabilities. Our analysis showed that all models were distinguishable, which confirms the validity of our model fitting pipeline (S3 Fig).

## Ex ante and ex post model simulations

We conducted ex ante model simulations to confirm and preview the value-updating and world-updating models' distinct predictions on the stimulus-dependent feedback effects under the current experimental setting. Model simulations were conducted using trial sequences (i.e., stimulus order and correct answers) identical to those administered to human participants. The model parameters used in the ex ante simulation are summarized in the Table A in S1 Appendix. Note that the 25 levels (uniformly spaced [0.15, 3.27]) of $\sigma_m$, the only parameter common to the 2 models, were used. As for the other parameters specific to each model, we selected the values that generated human-level task performances (see S4 Fig for details and statistical results). Simulations were repeated 100 times, resulting in the $100 \times N \times M \times K = 507,300 \sim 510,000$ trials per participant. For simplicity, we assumed neither lapse trials nor any arbitrary choice bias.

The procedure of ex post model simulations was identical to that of ex ante model simulations except that the best-fitting model parameters and lapse trials were used.

## Statistical tests

Unless otherwise mentioned, the statistical comparisons were performed using paired *t* tests (two-tailed, $N = 30$). To test the reversed feedback effects under conditions of strong sensory evidence, we applied one-sample *t* tests (one-tailed, $N = 27$ for S7A Fig, $N = 8$ for S7B Fig). Repeated *t* tests on PSEs between data and model (Figs 7B, 7C and S5) were performed (two-tailed, $N = 30$). In Table D in S1 Appendix, we reported the number of test conditions of significant deviation from the data (Bonferroni-corrected threshold; *: $P < 0.00083$, **: $P < 0.000167$, ***: $P < 0.0000167$). Additionally, Wilcoxon signed-rank tests were performed with the same threshold applied (Table D in S1 Appendix). Repeated *t* tests on each cell of episode frequency maps between the data and the models (Figs 7E, 7F and S6) were performed, and *P* values were subjected to Bonferroni correction (two-tailed, $N = 30$; value-updating, $P < 0.0000631$; world-updating, $P < 0.0000633$). Task performances between human agents ($N = 30$) and model agents with different sets of parameters ($N = 25$) were compared based on unpaired *t* tests (two-tailed, S4 Fig).

## Supporting information

**S1 Fig. Schematic illustration of BMBU's account of how the joint contribution of the sensory and feedback evidence to boundary updating leads to the reversal of choice bias as a function of sensory evidence strength. (A)** Reversal of subsequent choice bias—expressed in PSE—as a function of sensory evidence strength and boundary inference—expressed in likelihood computation—based on a PDM episode. Left panel: The circles with different colors (indicated by (b-d), which points to the corresponding panels below **(B-D)**) represent the PSEs associated with the boundary updating for 3 example PDM episodes, where the stimulus ($S_t$) varies from 0 to 2 while the choice ($C_t$) and feedback ($F_t$) are *large* and *correct*, respectively. Right panel: At the core of boundary inference is the computation of the likelihood of the class boundary based on the mnemonic measurement ($m'_t$) and the informed state of the class variable ($CL_t$), where $CL_t$ is jointly determined by $F_t$ and $C_t$ (see **Materials and methods** for the full computation of boundary inference in BMBU). **(B-D)** The likelihoods of the class boundary given the 3 example PDM episodes defined in **(A)**, where sensory evidence varies from the low **(B)**, to the intermediate **(C)**, and to the high **(D)** level. To help understand why and how, given the same feedback evidence, the direction of boundary updating reverses as the sensory evidence strengthens, we visualize the boundary likelihoods as a product of 2 functions (Eq 12), indicated by subpanels marked as (1) and (2). Top row: As indicated by (1), we plot each boundary likelihood when only the mnemonic measurement is considered, assuming that no feedback is provided. Note that these likelihood functions are centered around the values of $m'_t$, by attracting the class boundary toward themselves, driving a shift towards the *large* side (i.e., positive side on the boundary axis). Middle-Bottom rows: When the feedback evidence is given—i.e., when the informed state of $CL_t$ is revealed as *large*—in addition to the mnemonic measurement, an additional piece of information about the class boundary arises. As indicated by (1) × (2), we plot each boundary likelihood (defined in **(A)**). As indicated by (2), we plot each function (Middle row), as the result of (Bottom row) divided by (Top row). The complementary cumulative distribution functions shown here are also centered around $m'_t$ because the *large* state of $CL_t$ means that the class boundary is located somewhere smaller than $m'_t$. Note that these skewed distributions push the inferred class boundary away from the state of $CL_t$ informed by feedback, driving a shift towards the *small* side (i.e., negative side on the boundary axis). Consequently, the influences from the sensory evidence and the feedback evidence counteract each other (Bottom row). Note that the likelihood functions are centered in the *small* side when the sensory evidence is weak **(B)**, in the neutral side when intermediate **(C)**, and in the large side when strong **(D)**. These systematic shifts of the class-boundary likelihood as a function of the strength of sensory evidence predict that the PSE of the psychometric curve for the subsequent trial ($t$+1) reverses its sign from negative to positive as a function of the stimulus size, as shown in **(A)**.
(TIF)

**S2 Fig. Example trial courses of estimated class boundary. (A)** An example trial history to show how a temporal trajectory of the class boundary inferred by BMBU. For example, at trial #1 (x-axis), a physical stimulus (symbol x) was 0, a sensory measurement (symbol o) was a positive value when the boundary belief (solid black bar; y-axis) was centered at 0. BMBU's choice was *large* (symbol square on the top of y-axis), and correct feedback (same square filled with green color) was provided, which indicates that the class variable at trial #1 $CL_1$ was *large* (arrow's direction indicates the effect of the trial class variable on the subsequent boundary-updating). BMBU updates one's belief based on evidence from stimulus (colored symbol o) and feedback ($CL_1$), available at the time of boundary-updating. To illustrate cases where the bias reversal we defined in Fig 3D in the main text happen and do not happen, same examples

were intentionally used as those we used in S1 Fig where we further detailed on the model's mechanisms. Depending on colors, sensory evidence is weak (yellow symbol o) or strong (purple symbol o), which leads to whether or not the reversal happens. Trial cases featured in a red box indicates that the "Reinforcement" principle is held (predicting subsequent choices to repeat *large* choice) while those featured in a green box indicates that the "Reversal" happens (predicting subsequent choices to reverse the previously made *large* choice). **(B)** Temporal trajectories of the class boundary when the same 6-trial sequence of physical stimuli in **(A)** was simulated for 100 times. This means different $m$ and $m'$ were realized. The data underlying this figure **(A, B)** can be found in S1 Data.
(TIF)

**S3 Fig. Model recovery analysis.** Each square represents exceedance probability $p_{exc}$ from model recovery procedure. The "ground-truth" model to simulate synthetic behavior was correctly recovered with $p_{exc} > 0.9$ for all 4 models considered in the study. The light shade of the diagonal squares indicates that the ground-truth model was the best-fitting model, leading to a successful model recovery. Numerical values can also be found in S1 Data.
(TIF)

**S4 Fig. Histograms of classification accuracies of the human participants and their model partners in the ex ante simulations. (A, B)** Across-individual distributions of the classification accuracy of the belief-based RL model **(A)** and BMBU **(B)** overlaid on those of the human participants. The models' choices were generated via ex ante simulations with a specific set of model parameters (Table A in S1 Appendix), the results of which are depicted in Figs 4 and 5. The classification accuracy is measured by calculating the percentage of the trials in which the choice matched the feedback used in the actual experiment. The empty bars correspond to the histogram of human performances, the range of which is demarcated by the dashed vertical lines ([min, max] = [60.65%, 73.94%]). The average human classification accuracy was 67.85%. **(A)** Comparison of classification accuracy between the belief-based RL model's simulation (red color) and the human choices. The model's ex ante simulation accuracy was not different from the human accuracy ($t(53) = 1.4429$, $P = 0.1549$; Null hypothesis: model's performance vector and humans' performance vector come from populations with equal means, unpaired two-tailed $t$ test). **(B)** Comparison of classification accuracy between BMBU's simulation (green color) and the human choices. The model's ex ante simulation accuracy was not different from the human accuracy ($t(53) = 0.9707$, $P = 0.3361$, unpaired two-tailed $t$ test). There was no significant difference in classification accuracy between the value-updating model and BMBU ($t(48) = 0.5733$, $P = 0.5691$, unpaired two-tailed $t$ test). The data underlying this figure **(A, B)** can be found in S1 Data.
(TIF)

**S5 Fig. Retrospective (left columns), prospective (middle columns), and subtractive (right columns) history effects in PSE for the "Hybrid" model's ex post model simulations.** Top and bottom rows in each panel show the PSEs associated with the *toi* episodes involving *correct* and *incorrect* feedback at *toi*. Symbols with error bars, mean ± SEM across the 30 model agents, which correspond to their 30 human partners. The colors of the symbols and lines label choices (blue: *small* and yellow: *large*). The data underlying this figure can be found in S1 Data.
(TIF)

**S6 Fig. Maps of frequency deviations of the value-updating (A) and world-updating (B) model agents' classifications in the ex post simulations from the human decision-makers in the retrospective (left) and prospective (right) history effects.** Each cell represents a pair

of PDM episodes, as specified by the column and row labels. At each cell, the color represents how much the episode frequency observed in the model agents deviates from that observed in the corresponding human decision-makers. The results of statistical tests on these deviations are summarized in Fig 7E and 7F. The data underlying this figure (**A, B**) can be found in S1 Data.
(TIF)

**S7 Fig.  Retrospective (left columns), prospective (middle columns), and subtractive (right columns) history effects in PSE for the human classification performances of Urai and colleagues' work** [37] **(A) and Hachen and colleagues' work** [31] **(B). (A, B)** We downloaded both publicly available datasets, analyzed them in the same way that we analyzed human observers in our work, and plotted the results in the same format used for Fig 7A. Top and bottom rows in each panel show the PSEs associated with the *toi* episodes involving *correct* and *incorrect* feedback. Symbols with error bars, mean ± SEM across human observers. The colors of the symbols and lines label choices (blue: *small* and yellow: *large*). The overall patterns of the PSEs plotted here appear similar to those plotted in Fig 7A, displaying the reversals in direction of stimulus-dependent feedback effects. When the same statistical tests used in our work were carried out, some of the data points at the stimuli with strong sensory evidence at *toi* significantly deviated from zero in the direction opposite to the feedback effect predicted by the value-updating scenario, as indicated by the asterisks. **(A)** Sequential features of human observers ($N = 27$) analyzed in our way from human dataset that once had been published [37], which is openly available (http://dx.doi.org/10.6084/m9.figshare.4300043), then analyzed in the previous study [9]. In this study, the participants performed a binary classification task on the difference in motion coherence by sorting the pairs of random-dot-kinematogram stimuli shown in 2 intervals (s1 and s2) into one of the 2 classes ("s1<s2" vs. "s1>s2") over consecutive trials. The presented stimuli were taken from 3 sets of difficulty levels (the difference between motion coherence of the test and the reference stimulus; easy: [2.5, 5, 10, 20, 30], medium: [1.25, 2.5, 5, 10, 30], hard: [0.625, 1.25, 2.5, 5, 20]). As done in the original study [9], we binned the trials into 8 levels by merging the trials of 2 neighboring coherence levels (e.g., the coherence levels of [0.625, 1.25]) into a single bin. Note that the coherence bins of [20, 35, 45, 48.75, 51.25, 55, 65, 80] (%s1) on the x-axis (50% represents the equal coherence between s1 and s2) are matched to the x-axis in Fig 8 of the previous study in which the same dataset had been used. Asterisks mark the significance of one-sample *t* tests (uncorrected $P < 0.05$, one-tailed in the direction of feedback effects) on the panel *toi*+1 (stimulus 80%: $t(26) = 2.0138, P = 0.0272$) and on the panel *subtracted* (stimulus 20%: $t(26) = -3.1900, P = 0.0018$, stimulus 80%: $t(26) = 3.8810, P = 0.0003$). **(B)** Sequential features of human observers ($N = 8$) published in another previous study [31]. We used the human dataset openly available as part of the repository (https://osf.io/hux4n). In this study, the participants performed a binary classification task on the speed of vibrotactile stimuli by classifying the speed of the presented vibration as "low-speed (weak)" or "high-speed (strong)." Note that the 9-level stimuli of [−4, −3,−2,−1,0,1,2,3,4] on the x-axis followed how data were encoded by the original study [31]. Asterisks mark the significance of one-sample *t* tests (uncorrected $P < 0.05$, one-tailed in the direction of feedback effects) on the panel *toi*+1 (stimulus −4: $t(7) = -3.6757, P = 0.004$, stimulus −3: $t(7) = -3.5252, P = 0.0048$, and stimulus −2: $t(7) = -2.0325, P = 0.04$) and on the panel *subtracted* (stimulus −4: $t(7) = -1.9848, P = 0.044$). The data underlying this figure (**A, B**) can be found in S1 Data.
(TIF)

**S1 Appendix. Supporting details.** Supplemental details (Text) on additional model specifications of BMBU are provided. Supplementary tables (A-D Tables) to support the Results section

are provided. Table A. Parameters used for ex ante simulations. Table B. Parameters recovered from fitting the main models, world-updating and value-updating models, to human choices ($N$ = 30). Table C. Parameters recovered from fitting the rest of the models to human choices ($N$ = 30). Table D. Statistical results on model behavior versus human behavior in terms of PSE measures.
(DOCX)

**S1 Data. Excel spreadsheet containing, in separate sheets, the underlying numerical data for Figs** 2D, 4B, 4D, 4E, 4G, 4H, 5B, 5D, 5E, 5G, 5H, 6B, 6C, 6D, 7A, 7B, 7C, 7D, 7E, 7F, S2A, S2B, S3, S4A, S4B, S5, S6A, S6B, S7A and S7B.
(XLSX)

**S2 Data. Excel spreadsheet containing detailed statistical information comparing alternative PSE estimation methods.**
(XLSX)

## Acknowledgments

The authors are grateful to Daeyeol Lee for his insightful comments and inspiring conversations concerning the prior version of the manuscript.

## Author Contributions

**Conceptualization:** Hyang-Jung Lee, Heeseung Lee, Sang-Hun Lee.

**Data curation:** Issac Rhim.

**Formal analysis:** Hyang-Jung Lee.

**Funding acquisition:** Sang-Hun Lee.

**Investigation:** Hyang-Jung Lee.

**Methodology:** Hyang-Jung Lee, Chae Young Lim, Sang-Hun Lee.

**Resources:** Sang-Hun Lee.

**Software:** Hyang-Jung Lee.

**Supervision:** Sang-Hun Lee.

**Validation:** Hyang-Jung Lee, Chae Young Lim, Sang-Hun Lee.

**Visualization:** Hyang-Jung Lee, Sang-Hun Lee.

**Writing – original draft:** Hyang-Jung Lee.

**Writing – review & editing:** Hyang-Jung Lee, Heeseung Lee, Chae Young Lim, Issac Rhim, Sang-Hun Lee.

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
