## [Editor Report · Decision Letter 0]

27 Feb 2023

Dear Dr Lee, 

Thank you for submitting your manuscript entitled "What humans learn from corrective feedback for perceptual decision-making" for consideration as a Research Article by PLOS Biology.

Your manuscript has now been evaluated by the PLOS Biology editorial staff as well as by an academic editor with relevant expertise, and I am writing to let you know that we would like to send your submission out for external peer review.

Once your full submission is complete, your paper will undergo a series of checks in preparation for peer review. After your manuscript has passed the checks it will be sent out for review. To provide the metadata for your submission, please Login to Editorial Manager (https://www.editorialmanager.com/pbiology) within two working days, i.e. by Mar 01 2023 11:59PM.

Kind regards,

Lucas

Lucas Smith, Ph.D.

Associate Editor

PLOS Biology

lsmith@plos.org

---

## [Decision Letter · Decision Letter 1]

4 May 2023

Dear Dr Lee,

Thank you for your patience while your manuscript "What humans learn from corrective feedback for perceptual decision-making" was peer-reviewed at PLOS Biology, and I apologize again for our delay in sending you a decision. Your study has now been evaluated by the PLOS Biology editors, an Academic Editor with relevant expertise, and by several independent reviewers. In light of the reviews, which you will find at the end of this email, we would like to invite you to revise the work to thoroughly address the reviewers' reports.

As you will see below, the reviewers find the study interesting, however Reviewers 1 and 3 make clear suggestions for additional data presentation and analyses to strengthen the study. We think their requests are reasonable and we strongly encourage you to thoroughly address their comments. We think that some of reviewer 3's comments highlight that the results would still be interesting even if all traces of reinforcement-guided learning were not absent; s/he is pointing out that even a combination of world model inference and reinforcement-guided learning is still interesting. Given R3’s constructive comments, we think it might be especially useful to consider performing the additional tests s/he suggests in the manner they suggest.

Given the extent of revision needed, we cannot make a decision about publication until we have seen the revised manuscript and your response to the reviewers' comments. Your revised manuscript is likely to be sent for further evaluation by all or a subset of the reviewers.

**IMPORTANT - SUBMITTING YOUR REVISION**

*Re-submission Checklist*

*Published Peer Review*

*PLOS Data Policy*

*Blot and Gel Data Policy*

Sincerely,

Lucas

Lucas Smith, Ph.D.

Associate Editor

PLOS Biology

lsmith@plos.org

REVIEWS:

Reviewer #1: The authors investigate the origins of correcting feedback in perceptual decision making by using computational modelling and human experiments. When comparing the predictions of a value-based model vs a world's inference model, they find that the second type explains the data much better. Most importantly, the two models different by the type of qualitative predictions that they make: the world's inference model is the only one that can explain a bias reversal that happens as a function of stimulus strength. The human experiment results are intriguing, and the model seems to capture very fine details of the experiment.

Although I like the paper, I think that there are some aspects of it that deserve deep attention by the authors.

First, humans are faced with 5 stimulus values, but the model does not seem to use this fact at any point (Is the distribution over stimulus values assumed to be a discrete?). To be honest, I am not able to see what the prior distribution over the biases (B) is, or how it evolves over trials (no example is provided). This is important, because inference about the mean of the distribution will strongly depend on the assumed distribution over the stimulus strengths that the participants see. In particular, how does it affect the fact that there is a maximum strength value that can be ever observed? My intuition says that the model does not take this into account, but then I would not be sure about whether the bias reverses with stimulus strength in a model where inference is based on the knowledge that there is a maximum stimulus strength. In general, the description of the model is quite confusing, mostly because of the lack of explicit equations for some basic quantities. 

If there are only 5 stimulus values, why participants not just memorize them and simplify the computation of what is large and small? In particular, confidence would be very low when stimulus strength is zero, but this is not tested in the experiments, nor in the model. 

Second, I wonder how general the results are. In many PDM tasks, the boundary is very natural and intuitive: a vertical line boundary and detect whether stimulus is moving to the right or to the left respect to it. Is there any evidence that boundaries that hold some symmetries are also subject to learning?

I could not see any difference between Figs. 4 and 5. Are they identical?

Several important works on slow fluctuations and role of feedback in biases and behavior are missing and are relevant for the introduction and discussion:

Prefrontal cortex represents heuristics that shape choice bias and its integration into future behavior - ScienceDirect

https://www.nature.com/articles/ncomms14823

Given that the number of parameters is not very large compared to the number of trials, it is unlikely that adding lapses in the fitting would introduce overfitting, which means that it would be desirable to use lapses rate to better estimate biases and avoid any possible confound. 

I do not fully understand the role of m' and m. As m is both hidden to the experimenter and to the agent, I would recommend removing it from the model, unless I am missing something important that makes both of them necessary. 

The inequalities in Eq. of line 648 are very tricky, as the stimulus strengths are discrete. What happens when B=0? 

Eq of Line 654, I guess that CL here refers only to "small". Otherwise, I do not see how it would be correct. 

Line 657. "for simplicity" no, you define quantities in this way.

Eq of 661 is not identical to the previous equation in line 655? 

Minor:

Line 49: "The RL algorithm". RL refers to a set of algorithms, not to a single algorithm.

Line 87. Unclear what the "stronger stimulus strength" means. Later it is used "for strong evidence", but all that seems to contradict each of the sentences. Please, clarify. Overall, that passage is not 

very clear, and more intuition might be needed. 

Line 237. Clarify that "choice" refers to the "next choices"

Line 238 and following one. It would help to clarify how changes of B are related to changes of PSE. 

Line 239. It is unclear what "the size of 0 will hardly move…" means.

Line 629. This last sentence is repetitive with respect to the next one. 

Reviewer #2: In this paper the authors seek to determine whether models that cast perceptual choice updating in response to feedback as a process of learning the value associated with each alternative as opposed to learning the statistics of the environment. This is an interesting question because it touches on the fact that while animals learn perceptual decision tasks through a process of reinforcement learning, humans are trained in a very different way. Behavioural data were collected from participants performing a task in which rings had to be classed as larger or smaller than the average observed distribution. The authors hypothesise that the two different models make distinct predictions for the influence of feedback on subsequent choices. Where the value-learning models would predict that 'correct' feedback following a stimulus that is far from the currently perceived category boundary will increase the probability of selecting that alternative on subsequent trials, the world statistic model makes the opposite prediction. The authors find that a bayesian implementation of the world statistics model provides a markedly superior fit to the data compared to the value-based learning model. 

I think this paper can make a potentially important contribution to the literature however I am not sufficiently expert in either reinforcement learning models or bayesian modelling to evaluate the methodology. I do wonder whether the paper would be more suited to PLOS Computational Biology given the complexity of the model comparisons. 

My one comment is that in the introduction the authors should be more clear about what is meant by 'strong sensory evidence' - do they mean a larger ring/tree? or a ring/tree that is further from the current choice boundary irrespective of size? I believe it's the former. 

Reviewer #3: In a perceptual decision-making task, Lee at al. study the behavior of human subjects and show that they use evidence to update a "world model", thereby learning some sort of underlying model of the world. They bring some evidence that human behavior doesn't conform to a more popular RL-based account, which learn values from trial-and-error.

The article is well thought out: it presents an interesting question (whether humans use the statistical structure of the world for behavior) based on previously observed counter-intuitive behavioral signatures. Lee et al. show that the "world model" alternative can explain away these signatures. A big strength of the paper is the simulation analysis which enables us to understand why assuming a statistical structure allows for a better explanation of the human behavior.

However, this paper suffers from a serious limitation: whereas I am convinced from this work that the human behavior uses some sort of "world model", I am not convinced that "human decision-maker do not conform to [the reward-based decision-making] assumption". I do not see why the human's internal model could not be a mixture of both. Importantly, it does not have to be implemented in the brain as a mixture of strategies to be better explained in modelling by a mixture of strategies. I see that the authors have tried to address this concern by performing a model comparison using a "hybrid" model as a candidate model. However, I found this to be rather unconvincing for 2 reasons: 1/ why isn't the "hybrid" model included in the simulation analysis? Only the Value and World models are included figure 7 even though the "hybrid" model is the second-best performing model (with inconclusive evidence for the AIC). 2/ the model comparison between the "World" and "Hybrid" model is rather unclear, BIC is well-known to over-penalize models with more parameters, so it is not very surprising that "hybrid" model is going to lose in the terms of BIC (in the sense that the choice of the criteria penalizes the "hybrid" model).

Generally speaking, if you use BIC and AIC as criteria, one:

- should use the BIC if you want the model with the most parameters to win (the BIC favors the model with least parameters)

- one should use the AIC if you want the model with the least parameters to win (the AIC does not favor the model with least parameters)

Here, we are in the latter case and AIC gives inconclusive results for the comparison "world" against "hybrid" models.

When using AIC or BIC criteria, it is also important to perform some model recovery procedures to verify that the models are dissociable given the experimental task, the fitting procedure and the comparison metric. Two things: 1/ if the model recovery procedure shows that the models are dissociable with the BIC criteria, then the model selection argument based on BIC would be sound. And 2/ to avoid the issue of having to choose between AIC an BIC, it would be preferable to use a cross validation procedure. Holding out some sequences, fitting on the held-in sequences and testing on the held-out. This would provide a stronger argument for the world model over the hybrid model.

Also, to make this claim (world model over mixture model), it would be important to try a simpler alternative of a mixture of both strategies: (1 - w) world + w value (instead of assuming a multiplication - page 7). Likewise, to rule out RL models, it would be important to test for RL-based alternatives that tend to work better on human behavior, such as Pearce-Hall (learning rate scaling with absolute prediction error).

If these extra tests are not performed, the authors would have to tone down the claim against the "reward-based decision-making" assumption and make it more about the presence of "statistical structure" in the behavioral strategies (rather than the absence of RL strategies)

Figure 4 and 5 are identical (or at least I think) - I spent 30min trying to find differences. Am I missing something? 

Minor:

Equations in the methods are confusing. For instance, the integrals presented in the methods are integrals over probability laws (so we have an integral over a Gaussian law), I guess that it means that I should consider the probability density function and not the law but when I did that, I still didn't fully understand all the equations (and it requires efforts on the reader's side). I would recommend that the authors re-write the methods as formally as possible. 

The paragraph that introduces the "stimulus-dependent bias reversal" (last paragraph of page 9) is really hard to understand and seems to be an important paragraph. I had to look into the equations in the methods to grasp the concept.

Quality of figures were bad so there are some things that were hard to decipher (e.g. figure 4F)

Are \\sigma_m and \\sigma_m' identifiable? I can understand that they have different "meanings" in the model but in the fitting procedure it seems like we are adding an unnecessary parameters and the only thing that we can fit is (\\sigma_m)^2 + (\\sigma_m')^2.

Why is the \\sigma_diffusion parameters necessary? It makes the inference over B sub-optimal, what happens if you don't assume it and you assume in the exact inference for B?

---

## [Decision Letter · Decision Letter 2]

20 Sep 2023

Dear Dr Lee,

Thank you for your patience while we considered your revised manuscript "What humans learn from corrective feedback for perceptual decision-making" for publication as a Research Article at PLOS Biology. This revised version of your manuscript has been evaluated by the PLOS Biology editors, the Academic Editor and by two of the original reviewers. 

Based on the reviews, we are likely to accept this manuscript for publication, provided you satisfactorily address the remaining point raised by the reviewer 3.

**IMPORTANT: As you address reviewer 3's comments, please also make sure to address the following data and other policy-related requests:

1) TITLE: After some discussion within the team, we think the title should be edited to reflect some of the specific findings of your study more clearly. If you agree, we suggest you change it to something like:

"During perceptual decision-making humans treat corrective feedback as evidence of the state of the world rather than as a reward"

2) ABSTRACT: We think the abstract is currently written for a somewhat specialized audience of experts working on this topic. With our broad readership in mind, we encourage you to edit the abstract to make the findings more accessible to a general audience. It may be helpful to send your abstract to a colleague who does not work right in this field, for feedback. In the introduction, please spell out PDM when it is first used. 

3) ETHICS STATEMENT: Thank you for providing an ethics statement. Can you please update it to note whether this study has been conducted according to the principles expressed in the Declaration of Helsinki or some other national/international guidelines?

4) DATA and CODE: I see that your data availability statement says "All data and codes will be only available after acceptance in a public repository" - but I could not find a link to access this data anywhere (sorry if I missed it). 

PLOS has a data, which requires that all data underlying the study be made available without restriction: http://journals.plos.org/plosbiology/s/data-availability. Similarly, we will require that you provide any code that you have generated to support the conclusions of your manuscript, without restrictions upon publication. 

>>Please, can you provide me with the relevant repository links so that I can make sure your data and code meet our requirements? Please ensure that the code is sufficiently well documented and reusable and that your data has a readme file explaining what the data is and how it relates the the figures in your study. 

>>Please also ensure that figure legends in your manuscript include information on where the underlying data can be found. To each figure legend (including supplemental) you can add the sentence "the data underlying this figure can be found at ___" (and the reference the relevant data deposition). 

>>Please update your Data Statement in the submission system accurately describes where your data and code can be found.

We expect to receive your revised manuscript within two weeks. 

*Published Peer Review History*

*Press*

Sincerely,

Luke

Lucas Smith, Ph.D.

Senior Editor,

lsmith@plos.org,

PLOS Biology

Reviewer Comments: 

Reviewer #1: The authors have fully addressed all my comments.

Reviewer #3: 

Thank you for addressing my concerns and providing a detailed response to my comments. I appreciate your clarification regarding the main claim of your work and how it relates to the goodness-of-fit results and the simulation analyses.

I also appreciate your efforts to improve the methodological rigor of your study, including using only the AIC criteria, performing a model recovery analysis, and presenting the ex post simulation results for the Hybrid model. These changes strengthen your paper and the robustness of your findings.

Regarding the suggestion for an alternative variant of the Hybrid model, I understand your explanation for why it cannot be implemented in your common platform. Given this limitation, it's reasonable to focus on the Hybrid model as defined in your approach.

Overall, your response has addressed my concerns effectively, and the revisions you've made to the manuscript enhanced the quality and clarity of your work.

One overall remark: every time a t-test is performed, the tvalue and number of degrees of freedom should be systematically reported (which is not the case here, e.g., see the "Evaluating the two scenarios for the goodness of fit to human decision-making data")

---

## [Editor Report · Decision Letter 3]

10 Oct 2023

Dear Sang-Hun,

Thank you for the submission of your revised Research Article "Corrective feedback guides human perceptual decision-making by informing about the world state rather than rewarding its choice" for publication in PLOS Biology, and thank you for addressing the last reviewer and editorial requests in this revision. On behalf of my colleagues and the Academic Editor, Matthew F. S. Rushworth, I am pleased to say that we can in principle accept your manuscript for publication, provided you address any remaining formatting and reporting issues. These will be detailed in an email you should receive within 2-3 business days from our colleagues in the journal operations team; no action is required from you until then. Please note that we will not be able to formally accept your manuscript and schedule it for publication until you have completed any requested changes.

**Please note that following our conversation over email, I have updated the data availability statement for your manuscript to include the DOI that you provided for your github deposition. Please do double check that everything looks good to you after this change. 

PRESS

Sincerely, 

Lucas Smith, Ph.D.

Senior Editor

PLOS Biology

lsmith@plos.org